# FAULT TOLERANT MULTI-AGENT LEARNING WITH ADVERSARIAL BUDGET CONSTRAINTS

## ABSTRACT

We study robustness to agent malfunctions in cooperative multi-agent reinforcement learning (MARL), a failure mode that is critical in practice yet underexplored in existing theory. We introduce MARTA, a plug-and-play robustness layer that augments standard MARL algorithms with a Switcher-Adversary mechanism which selectively induces malfunctions in performance-critical states. This formulation defines a fault-switching $(N + 2)$-player Markov game in which the Switcher chooses when and which agent fails, and the Adversary controls the resulting faulty behaviour via either random or worst-case policies. We develop a Q-learning-type scheme for this setting and show that the associated Bellman operator is a contraction. This yields existence and uniqueness of the minimax value, convergence to a Markov perfect equilibrium, and theoretical guarantees for both the base formulation and a budgeted variant (MARTA-B). Empirically, MARTA integrates seamlessly with MARL algorithms without architectural modification and consistently improves robustness under injected malfunctions across Traffic Junction, LBF and MPE SimpleTag. These results position MARTA as a theoretically grounded and practically deployable mechanism for fault-tolerant MARL.

## 1 INTRODUCTION

In multi-agent systems (MAS), interacting agents need to anticipate each other's actions and coordinate to perform tasks successfully (Albrecht et al., 2024). Critical to the success of agents solving tasks is a reliance on agents executing expected actions. This enables agents to coordinate with other agents whose actions may not be directly observable. To encourage learning coordinated behaviour, leading MARL methods perform centralised-training with decentralised execution to train MARL policies (Rashid et al., 2018; Oliehoek et al., 2016). In this paradigm, agents observe each other during training while using only local observations during execution. The information about the global system aids the agents to anticipate the actions of others during execution when only partial information about their surroundings is available. Despite the benefits to coordination, these aspects of MARL training leave MARL protocols vulnerable to catastrophic outcomes whenever agents suffer from malfunctions that cause unexpected behaviour. Nevertheless, malfunctions of computerised devices are commonplace (Cristian, 1991). For example, within factory environments, factory robots are often tasked with coordinating their actions with other agents. Failure to anticipate correct actions can prevent task completion and lead to dangerous outcomes. This extends to settings where a single agent is factorised into several sub-controllers, as in multi-agent MuJoCo or dexterous hand benchmarks. Here, each agent independently controls part of the automaton. This presents an urgent challenge: to devise MARL methods capable of performing in the presence of agent malfunctions.

In single agent reinforcement learning (RL), there is a rich literature on generating policies that are capable of robust performance in scenarios of failure, namely fault-tolerant (FT) policies (Mguni, 2019; Fan et al., 2021). A standard technique, adopted from control theory, is to introduce an adversarial agent that chooses actions to minimise the agent's expected return (Pinto et al., 2017). The adversary seeks to induce outcomes that correspond to worst-case scenarios thus generating experiences for the agent to learn a robust policy. This setup results in a zero-sum game between the adversary and the 'controller'. Owing to their structural properties, zero-sum games are particularly amenable to theoretical analysis (Osborne & Rubinstein, 1994). Nevertheless, the inclusion of

an adversary that executes actions in exact opposition to the agent's goals can induce extremely pessimistic behaviour, leading the controller to proceed with extreme caution when attempting to perform its task (Grau-Moya et al., 2018). This can undermine performance given a higher FT level. In MAS, small changes in an agent's behaviour due to malfunctions in the system can cause massive changes in the system behaviour. Each agent may change its behaviour in response to changes in other agents resulting in large cumulative deviations from expected outcomes (Steinberg & Zangwill, 1983). This phenomenon is studied within the context of game theory where it has been shown that equilibrium solutions of games can change violently as a result of very small changes in any aspect of the system (Stahl II, 1988). Consequently, coordination which is often a critical ingredient to MARL solutions, is liable to break down in the presence of malfunctions (Slumbers et al., 2023).

To tackle these challenges, we introduce a novel fault-tolerant MARL framework, MARTA, that enables MARL agents to robustly respond to malfunctions and optimally respond to the presence of failures while performing tasks. We consider in particular, team-reward scenarios in which agents may suffer malfunctions and cease to follow the policies they had learned to perform the task. MARTA includes an adaptive RL agent called Switcher that observes the *joint behaviour* of the agents and uses this information to determine malfunctions on its choice of agent that have the potential to disrupt coordination and induce the greatest harm to the system performance. To do this, MARTA employs *switching controls* (Øksendal & Sulem, 2007; Mguni et al., 2023) to enable Switcher to learn which states and agents, when malfunctioning, are most detrimental to the system. The malfunction actions are executed by a dedicated Adversary agent. In response, the active MARL agents learn how to best-respond to the malfunctioning agent. MARTA includes a parameter that calibrates the trade-off between fault tolerance and performance. It also has a budget facility that controls the number of attacks inflicted during training allowing for further calibration of its policies' FT levels. Since, MARTA is selective over the states it induces a fault and the choice of agent it addresses the challenge of how to respond against failures at critical system states. MARTA also makes use of information about the joint behaviour of agents to decide its adversarial attacks during training. This enables it to identify malfunctions that most disrupt coordination. Conceptually, MARTA reframes FT as a strategic interaction between three roles: cooperative agents seeking task success, an Adversary modelling agent failure, and a Switcher that decides when and where failures should occur. This design reflects real-world systems where failures are neither constant nor uniformly harmful, but selectively catastrophic when they arise at coordination-critical states. MARTA selectively induces state-dependent malfunctions during training, strengthening coordination without the conservatism of always-on adversaries.


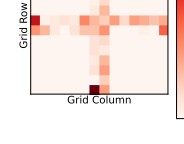

(a) Traffic Junction Map    (b) Switcher activation heatmap

We demonstrate this feature in the Traffic Junction environment Koul (2019). Here, we show the activation decisions made by Switcher at each timestep during evaluation. We then aggregate the activation frequency over the grid map to generate a heatmap, see Fig. 1(b). We observe that Switcher does not activate faults uniformly across space and the interventions are heavily concentrated at areas in which coordination is critical, namely at the intersections and entry/exit points that determine if the agents will collide and focussing on adversarial malfunctions at *critical control points* where a single failure can maximally disrupt traffic flow or induce collisions. Corners and side-entry positions where agents are most likely to merge into congested traffic, also exhibit elevated activation rates. Adding Switcher and Adversary agents with their own goals leads to a nonzero-sum Markov game (MG) (Fudenberg & Tirole, 1991) with $N + 2$ agents. Convergence in general Markov games is rare. MARTA achieves convergence via a switching-augmented Bellman contraction (Yang & Wang, 2020). Nevertheless, using a special set of features in MARTA's design, we prove MARTA converges to a solution in which it learns to induce MARL policies that are robust to worst-case fault scenarios.

**Contributions.** This paper makes the following contributions:

1. We introduce MARTA, a framework for fault-tolerant MARL based on a Switcher–Adversary mechanism that selectively induces agent malfunctions in coordination-critical states.

2. We formulate a new fault-switching MG with state-dependent, adversarially-selected malfunctions. We prove the existence and uniqueness of a minimax value and the convergence of Q-learning procedures for this game, including with linear function approximators and budgeted faults.

3. We show how MARTA can be implemented as a plug-and-play robustness layer on top of standard MARL algorithms such as QMIX and VDN, without changing their architectures.

4. We evaluate MARTA on Traffic Junction, Level-Based Foraging and MPE SimpleTag, showing consistent robustness gains and reduced failure rates across discrete and continuous control tasks, and across both random and worst-case malfunction regimes.

Together, these results establish MARTA as a principled and practical approach to training MARL policies that are robust to realistic agent malfunctions. At a high level, MARTA introduces a fault-switching game in which a Switcher activates malfunctions and an Adversary controls faulty behaviour. Our theoretical contribution shows that this augmented game is not merely well-defined but admits a unique and stable solution. Specifically, we prove existence and uniqueness of the minimax value and convergence of Q-learning, including under linear function approximation to extend this to convergence guarantees under linear function approximation and explicit malfunction budgets. This structure is fundamentally distinct from classical zero-sum Markov games, where adversarial dynamics alone do not encode budgeted, state-dependent mode-switching behaviour.

## 2 THE MARTA FRAMEWORK

A fully cooperative MAS is modelled by a decentralised-Markov decision process (Dec-MDP) (Deng et al., 2023) which is an augmented MDP involving a set of $N \geq 2$ agents denoted by $\mathcal{N}$ that independently decide actions to take which they do so simultaneously over many rounds. Formally, a dec-MDP is a tuple $\mathfrak{M} = \langle \mathcal{N}, \mathcal{S}, (\mathcal{A}_i)_{i \in \mathcal{N}}, P, \mathcal{R}, \gamma \rangle$ where $\mathcal{S}$ is the finite set of states, $\mathcal{A}_i$ is an action set for agent $i \in \mathcal{N}$ and $\mathcal{R} : \mathcal{S} \times \boldsymbol{\mathcal{A}} \to \mathcal{P}(D)$ is the reward function that all agents jointly seek to maximise where $D$ is a compact subset of $\mathbb{R}$ and $P : \mathcal{S} \times \boldsymbol{\mathcal{A}} \times \mathcal{S} \to [0,1]$ is the probability function describing the system dynamics where $\boldsymbol{\mathcal{A}} := \times_{i=1}^{N} \mathcal{A}_i$ and $\gamma \in [0,1)$ specifies the degree to which each agent's rewards are discounted over time. We consider a partially observable setting; given the system state $s_t \in \mathcal{S}$, each agent $i \in \mathcal{N}$ makes local observations $\tau_i^t = \mathcal{O}(s_t, i)$ where $\mathcal{O} : \mathcal{S} \times \mathcal{N} \to \mathcal{Z}_i$ is the observation function and $\mathcal{Z}_i$ is the set of local observations for agent $i$. Each agent $i \in \mathcal{N}$ samples its actions from a *Markov policy* $\pi_{i, \boldsymbol{\theta}_i} : \mathcal{Z}_i \times \mathcal{A}_i \to [0,1]$, which is parameterised by the vector $\boldsymbol{\theta}_i \in \mathbb{R}^d$ and each $\pi^i \in \Pi_i$ which is a compact Markov policy space. We abbreviate $\pi_{i, \boldsymbol{\theta}_i}$ is as $\pi_i$ and denote by $\boldsymbol{\mathcal{Z}} := \times_{i \in \mathcal{N}} \mathcal{Z}_i$ and $\boldsymbol{\Pi} := \times_{i \in \mathcal{N}} \Pi_i$. At time $t$ the system is in state $s_t \in \mathcal{S}$ and each agent $i \in \mathcal{N}$ takes an action $a_t^i \in \mathcal{A}_i$. The *joint action* $\boldsymbol{a}_t = (a_t^1, \dots, a_t^N) \in \boldsymbol{\mathcal{A}}$ produces a one-step reward $r_i \sim \mathcal{R}(s_t, \boldsymbol{a}_t)$ for agent $i \in \mathcal{N}$ and influences the next-state transition which is chosen according to $P$. Each agent's goal is to maximise its expected returns measured by its value function $v(s|\boldsymbol{\pi}) = \mathbb{E}_{\boldsymbol{\pi}} \left[ \sum_{t=0}^{\infty} \gamma^t \mathcal{R}(s_t, \boldsymbol{a}_t) | s_0 = s \right]$, where $\boldsymbol{\pi} = (\pi^i, \pi^{-i}) \in \boldsymbol{\Pi}$ and $-i$ denotes the tuple of agents excluding agent $i$.

MARTA introduces an additional RL agent, Switcher that decides to induce a malfunction in one of the $N$ agents during the training phase. Specifically, Switcher selects an agent to malfunction at which point, the agent's actions are decided by an Adversary agent's policy while the other agents execute their intended policy. In response, during training, the remaining agents learn how to respond to the behaviour of the malfunctioning agent and in so doing, learn how to respond to agent malfunctions within the collective. A key feature is that Switcher and Adversary are equipped with their own objective that aims to produce malfunctions that inflict the greatest possible harm to the system (during training) including faults that undermine coordination required to solve the task. **Design rationale.** MARTA is designed to capture two practically important classes of failures in multi-agent systems. First, we consider *random malfunction behaviour*, where the Adversary executes a random policy. This regime models sensor degradation or noisy actuators, where control channels are unreliable but not explicitly hostile. Second, we consider a *worst-case adversary*, trained to choose actions that maximally degrade collective performance. This regime captures catastrophic faults such as corrupted control modules or compromised local controllers. The Switcher learns a policy over *when* to trigger a malfunction and *which* agent is affected, so that the base MARL agents learn best-responses to both mild and severe failures. In this way, MARTA produces policies that are robust to realistic actuator failures rather than only to abstract observation or action perturbations.

In MARTA, the policies $\boldsymbol{\sigma} = (\sigma^1, \dots, \sigma^N)$ and $\boldsymbol{\pi}$ first each propose the joint actions $\boldsymbol{a} \in \boldsymbol{\mathcal{A}}$ and $\boldsymbol{f} = (f^1, \dots, f^N) \in \boldsymbol{\mathcal{A}}$ respectively. These joint actions are then observed by the policy $\mathfrak{g}$. Switcher samples a discrete decision $g_t \in \mathcal{N}$ from its policy $\mathfrak{g}$. If $g_t = i \in \mathcal{N}$, then agent $i$ is selected to malfunction whereby its action is overridden by an Adversary policy $\sigma^i \in \Pi^i$ while the agents $-i$ sample their actions from their intended policy $\pi^{-i}$ i.e. the joint action $\boldsymbol{a}_t = (f_t^i, a_t^{-i}) \sim (\sigma^i, \pi^{-i})$ is executed in the environment. During training in MARTA, the $N$ agents seek to jointly maximise the following objective: $v(s|\boldsymbol{\pi}, \mathfrak{g}, \boldsymbol{\sigma}) = \mathbb{E}_{\boldsymbol{\pi}, \mathfrak{g}, \boldsymbol{\sigma}} [\sum_{t=0}^{\infty} \gamma^t \mathcal{R}(s_t, \boldsymbol{a}_t)|s = s_0]$. The Adversary and Switcher's objectives are $-v$ which captures their goal to find malfunctions that have the potential to induce the greatest harm to the system. Therefore, by learning an optimal $\mathfrak{g}$, Switcher acquires the optimal policy for activating Adversary. We later show in Theorem 2 that in response, the MARL agents in turn learn how to best-respond to such failures. As remarked earlier, we consider two scenarios for a malfunctioning agent $i$. The first scenario is when the Adversary action is sampled from a purely random policy and secondly, a worst-case action scenario.

**Switching Control Mechanism.** So far Switcher's problem involves learning to induce a malfunction by a single agent at *every* state. This can severely undermine performance, since agents must continually guard against large losses arising from adversarial actions at each decision step. Moreover, the current formulation restricts Switcher to selecting only one agent per episode, although in practice the critical fault may shift between agents as the system evolves. A more expressive approach allows Switcher to choose, at any state, both which agent malfunctions and whether to activate a malfunction at all. We therefore extend MARTA by equipping Switcher with *switching controls* (Mguni, 2018; Mguni et al., 2023). At each state $s$, Switcher makes a binary decision on whether to activate a malfunction for agent $i \in \mathcal{N}$. The policy $\mathfrak{g}$ determines whether a malfunction is induced and, if so, which agent's adversarial policy $\sigma^i$ is executed. To encourage selectivity, each activation incurs a positive cost $c > 0$, ensuring that malfunctions are only triggered when they cause a substantial reduction in expected system performance. The task of Switcher is therefore to learn a policy $\mathfrak{g}$ that activates adversarial malfunctions solely at states where they most degrade performance. For a given $\boldsymbol{\pi} \in \boldsymbol{\Pi}$, Switcher's objective is to find $\mathfrak{g}$ that *maximises*: [1]

$$v_S(s|\boldsymbol{\pi}, \mathfrak{g}, \boldsymbol{\sigma}) = -\mathbb{E}_{\boldsymbol{\pi}, \mathfrak{g}, \boldsymbol{\sigma}} \left[ \sum_{t=0}^{\infty} \gamma^t \left[ \mathcal{R}(s_t, \boldsymbol{a}_t) + c\mathbf{1}_{\mathcal{N}}(g(s_t)) \right] \right],$$

where $\mathbf{1}_{\mathcal{N}}(g) = 1$ if $g \in \mathcal{N}$ and $\mathbf{1}_{\mathcal{N}}(g) = 0$ otherwise. Switcher's action-value function is $Q_S(s, g|\boldsymbol{\pi}, \mathfrak{g}) = -\mathbb{E}_{\boldsymbol{\pi}, \mathfrak{g}} [\sum_{t=0}^{\infty} \gamma^t (\mathcal{R}(s_t, \boldsymbol{a}_t) + c \cdot \mathbf{1}_{\mathcal{N}}(g))|s_0 = s, g_0 = g]$. Adding the Switcher with an objective distinct from the $N$ agents results in a non-cooperative Markov game $\mathcal{G} = \langle (\mathcal{N}, \text{Switcher}, \text{Adversary}), \mathcal{S}, ((\mathcal{A}_i)_{i \in \mathcal{N}}, \mathcal{A}_S, \boldsymbol{\mathcal{A}}), P, (\mathcal{R}, \mathcal{R}_S, \mathcal{R}_A), \gamma \rangle$ where $\mathcal{A}_S := \{0\} \cup \mathcal{N}$ and $\mathcal{R}_S(s, \boldsymbol{a}, g) := -c \cdot \mathbf{1}_{\mathcal{N}}(g) - \mathcal{R}(s, \boldsymbol{a})$ denote Switcher agent's action set and its reward function respectively and $\mathcal{R}_A = -\mathcal{R}$ denotes Adversary's reward function. In MARL, having multiple learners with a payoff structure that is neither zero-sum nor a team game can occasion convergence issues (Shoham & Leyton-Brown, 2008). Moreover, unlike standard MARL, MARTA incorporates switching controls. Nevertheless, we prove MARTA converges under standard assumptions.

The parameter $c$ plays an important role in calibrating the fault-tolerance of the system of MARL agents. Larger values of $c$ incur higher costs for each malfunction making Switcher more selective about inflicting malfunctions i.e. limiting interventions to the states where the harm is greatest (this is proven in Theorem 1). In turn, the agents learn how to best-respond to only the most harmful malfunctions. In the limit $c \to 0$, we return to a classic robust framework where Switcher can profitably choose to inflict malfunctions at all states, leading to highly cautious joint policies which may harm performance for a given higher fault-tolerance. In Sec. 4, as an alternative to using the cost $c$, we study a setup with a budget constraint on the number of malfunctions Switcher can inflict.

## 2.1 DETAILS ON ARCHITECTURE

We now describe a concrete realisation of MARTA's core components which consist of $N$ MARL agents, an additional MARL agent called Adversary and a switching control RL algorithm as Switcher.

---

[1] We have employed the shorthand $\boldsymbol{a}_t \sim (\boldsymbol{\pi}, \boldsymbol{\sigma})$ to denote $\boldsymbol{a}_t = (a_t^j, a_t^{-j}) \sim \boldsymbol{\pi}$ when $g_t = 0$ and $\boldsymbol{a}_t = (f_t^j, a_t^{-j}) \sim (\sigma^j, \pi^{-j})$ when $g_t = j \in \mathcal{N}$ i.e. $v_S(s|\boldsymbol{\pi}, \mathfrak{g}, \boldsymbol{\sigma}) = -\mathbb{E}_{\boldsymbol{\pi}, \mathfrak{g}, \boldsymbol{\sigma}}[\sum_{t=0}^{\infty} \sum_{j \in \mathcal{N}} \gamma^t [(\mathcal{R}(s_t, (f_t^j, a_t^{-j})) + c) \mathbf{1}_{(g(s_t)=j)} + \mathcal{R}(s_t, \boldsymbol{a}_t)(1 - \mathbf{1}_{(g(s_t)=j)})]]$. For convenience we drop the dependence of $v_S$ on $\boldsymbol{\sigma}$.

Each (MA)RL component can be replaced by various other (MA)RL algorithms.

• $N$ **MARL agents**. Each agent has 2 policies: 1) an Action policy $\pi^i$ that aims to maximise the joint objective 2) Adversary policy $\sigma^i$ that aims to minimise the joint objective.

    ○ **Action policy:** We use QMIX (Rashid et al., 2018), an off-policy MARL value-based method that accommodates only action value functions to train the action policy.

    ○ **Adversary policy:** For training Adversary, we also use QMIX. The adversary selects actions to maximise $-Q = -\mathbb{E}_{\boldsymbol{\pi},\mathfrak{g}}\left[\sum_{t=0}^{\infty} \gamma^t \mathcal{R}(s_t, (f_t^i, \pi_t^{-i}))|s_0 = s, f_0^i = f^i\right]$.

**Role of the Adversary.** The Adversary defines the distribution over *faulted* actions that the system must be robust to. In the random-fault setting, $\sigma_i$ is a uniformly random policy, which serves as a proxy for stochastic sensor or actuator malfunctions. In the worst-case setting, $\sigma_i$ is trained to maximise $-Q$, deliberately inducing trajectories that are most damaging to the team objective. The Switcher then learns a policy $g$ that chooses *when* to hand control of agent $i$ to $\sigma_i$, exposing the cooperative agents to structured fault patterns during training.

• **Switcher Switching Policy** We use soft actor-critic (SAC) (Haarnoja et al., 2018) for Switcher policy. Its action set consists of $N + 1$ actions: 1) activate an agent $i$ Adversary 2) do not activate a malfunction. Switcher updates its policy while the MARL agents learn. The agents' actions are executed in the environment and the loop repeats. The trajectories generated by this process are stored in a replay buffer from which Switcher, the $N$ MARL agents, and Adversary are trained. Note that since at each state the decision space for Switcher is $\{0\} \times \mathcal{N}$ problem facing Switcher which is generally of relatively low complexity. This can result in a relatively quick training stage for Switcher relative to the base MARL learners. The pseudocode for MARTA is provided in the Appendix.

## 3 ANALYSIS OF MARTA

We now study the game induced by the MARTA, which is an $N + 2$ *non-cooperative Markov game* and prove the existence of a stable (equilibrium) solution where each agent enacts a policy that best responds to the system in the presence of (worst-case) malfunctions induced by Switcher.[2] We then show that the MARTA algorithm converges to the solution that both maximises Switcher agent's value function and the agents' joint objective. With this, Switcher agent learns to activate malfunctions only at the set of states at which doing so causes the highest overall reduction in the system performance. Meanwhile the $N$ agents learn to jointly best respond to such malfunctions. All results in this section are proven in the Appendix and are built under Assumptions 1 - 3 (Sec. D.1 of the Appendix) which are standard in RL (Bertsekas, 2012). Although MARTA is formulated as a Markov game, its structure substantially departs from classical zero-sum formulations Littman (1994). The fault-switching game introduces: (i) an explicit malfunction budget, (ii) state-dependent fault activation, (iii) persistent mode-switching dynamics, and (iv) a distinct Bellman operator incorporating switching costs and activation penalties. The contraction property proved in Lemmas 6-8 applies specifically to this switching-augmented operator and does not follow from generic Markov game results. This yields convergence behaviour not captured by standard adversarial learning frameworks.

We first observe that an optimal robust policy is one in which the $N$ agents play a best-response joint policy against Switcher who in turn executes a best-response policy. Our first task is to prove the existence of a solution to the game, namely an equilibrium stable point in which each agent enacts a best-response i.e. responds optimally to actions of other agents in the game $\mathcal{G}$. The following result establishes the existence and uniqueness of such a solution:

**Theorem 1.** *The minimax value of $\mathcal{G}$ exists and is unique, i.e. there exists a function $v^* : \mathcal{S} \to \mathbb{R}$ such that $v^*(s) := \min_{\hat{\mathfrak{g}}} \max_{\hat{\boldsymbol{\pi}} \in \boldsymbol{\Pi}} v(s|\hat{\boldsymbol{\pi}}, \hat{\mathfrak{g}}) = \max_{\hat{\boldsymbol{\pi}} \in \boldsymbol{\Pi}} \min_{\hat{\mathfrak{g}}} v(s|\hat{\boldsymbol{\pi}}, \hat{\mathfrak{g}}), \quad \forall s \in \mathcal{S}.$*

**Proof sketch.** The minimax equality and existence of $v^*$ follow from the contraction property of the induced Bellman operator. Specifically, Lemma 7 establishes monotonicity and Lemma 8 proves strict contraction under Assumptions 1–3 (Appendix D.1), implying a unique fixed point by the Banach fixed-point theorem. This fixed point coincides with the unique minimax value of the game.

An important consequence of Theorem 1 is that at the stable point, each of the $N$ agent best-respond to the influence of Switcher so that the agents jointly enact best-response a FT joint policy. This is captured formally in the following result:

---

[2]Note that since Adversary uses a stochastic policy, by considering the limiting variance case, the results also cover the scenario in which a malfunctioning agent executes random actions.

**Proposition 1.** *Let $\hat{\boldsymbol{\pi}} \in \boldsymbol{\Pi}$ be an equilibrium policy as defined in Theorem 1, then $\hat{\boldsymbol{\pi}}$ is a Markov perfect equilibrium[3] policy, that is to say, no agent can improve the team reward by responding to the malfunctions executed by Adversary with a change in their policy.*

Having established the existence of a solution to the game, we now turn to the question of how a set of MARL agents can learn such a solution. The following result proves the convergence of a dynamic programming procedure to the solution. It therefore lays the foundation for our MARL algorithm, MARTA which learns robust MARL policies in this setting. In particular, the following theorem proves the convergence of a MARTA to the solution $v^*$ by repeated application of a Bellman operator.

**Theorem 2.** *Let $v : \mathcal{S} \to \mathbb{R}$ then the sequence of Bellman operators acting on $v$ converges to the solution of the game, that is to say for any $s \in \mathcal{S}$ the following holds:*

$$\lim_{k \to \infty} T^k v(s|\boldsymbol{\pi}, \mathfrak{g}) = v^*(s), \tag{1}$$

*where $v^*(s) = \min_{\hat{\mathfrak{g}}} \max_{\hat{\boldsymbol{\pi}} \in \boldsymbol{\Pi}} v(s|\hat{\boldsymbol{\pi}}, \hat{\mathfrak{g}}) = \max_{\hat{\boldsymbol{\pi}} \in \boldsymbol{\Pi}} \min_{\hat{\mathfrak{g}}} v(s|\hat{\boldsymbol{\pi}}, \hat{\mathfrak{g}})$, the operator $T$ is given by $Tv(s|\boldsymbol{\pi}, \mathfrak{g}) = $*

$$\min \left\{ \hat{\boldsymbol{\mathcal{M}}} Q_S, \max_{\boldsymbol{a} \in \boldsymbol{\mathcal{A}}} \left[ \mathcal{R}(s, \boldsymbol{a}) + \gamma \sum_{s' \in \mathcal{S}} P(s'; \boldsymbol{a}, s) v(s'|\boldsymbol{\pi}, \mathfrak{g}) \right] \right\} \text{ where } \hat{\boldsymbol{\mathcal{M}}} Q_S(s, \boldsymbol{a}, g|\boldsymbol{\pi}, \mathfrak{g}) := c + $$
$$\min_{i \in \mathcal{N}, a^i \in \mathcal{A}^i} \max_{a^{-i} \in \mathcal{A}^{-i}} Q_S(s, (a^i, a^{-i}), g|\boldsymbol{\pi}, \mathfrak{g}).$$

An immediate consequence of the theorem is that MARTA learns to make the minimum number of malfunctions required to learn the solution to the agents' joint problem since any additional malfunctions would render Switcher agent's policy suboptimal. In Proposition-2 of the Appendix, we fully characterise Switcher's policy $\mathfrak{g}$, i.e. where Switcher should activate Adversary and which agent $i \in \mathcal{N}$ in terms of an obstacle condition which can be determined at each state. The following theorem shows that the system in which both the system of $N$ agents and Switcher agent train concurrently, the MARTA framework converges to the solution.

**Theorem 3.** *Consider the following Q-learning variant:*

$$\boldsymbol{Q}_{t+1}(s_t, \boldsymbol{a}_t, g) = \boldsymbol{Q}_t(s_t, \boldsymbol{a}_t, g)$$
$$+ \alpha_t(s_t, \boldsymbol{a}_t) \left[ \min \left\{ \hat{\boldsymbol{\mathcal{M}}} \boldsymbol{Q}_t(s_t, \boldsymbol{a}_t, g), \mathcal{R}_S(s, \boldsymbol{a}, g) + \gamma \max_{\boldsymbol{a}' \in \boldsymbol{\mathcal{A}}} \boldsymbol{Q}_t(s_{t+1}, \boldsymbol{a}', g) \right\} - \boldsymbol{Q}_t(s_t, \boldsymbol{a}_t, g) \right],$$

*then $\boldsymbol{Q}_t$ converges to $\boldsymbol{Q}^\star$ with probability 1, where $s_t, s_{t+1} \in \mathcal{S}$ and $\boldsymbol{a}_t \in \boldsymbol{\mathcal{A}}$.*

Theorem 3 proves the convergence of MARTA to the solution to the game $\mathcal{G}$ in which each agent optimally responds to the policies of other agents in the system. Function approximators enable parameterisation of key estimators in RL. Linear function approximators are an important class due to their simplicity and convexity (Jin et al., 2020). Moreover, they serve as an important stepping stone for proving analogous results with other function approximators. To this end, we now extend the convergence results to linear function approximations.

**Theorem 4.** *MARTA converges to the stable point of $\mathcal{G}$, moreover, given a set of linearly independent basis functions $\Phi = \{\phi_1, \ldots, \phi_p\}$ with $\phi_k \in L_2, \forall k$. Define a projection $\Pi$ on a function $\Lambda$ by: $\Pi \Lambda := \arg\min_{\bar{\Lambda} \in \{\Psi r | r \in \mathbb{R}^p\}} \|\bar{\Lambda} - \Lambda\|$. Then MARTA converges to a limit point $r^\star \in \mathbb{R}^p$ which is the unique solution to $\Pi \mathfrak{F}(\Phi r^\star) = \Phi r^\star$ where for any $\Lambda \in L_2$, $\mathfrak{F}\Lambda := \mathcal{R} + \gamma P \min\{\hat{\boldsymbol{\mathcal{M}}}\Lambda, \Lambda\}$ and $r^\star$ satisfies the following: $\|\Phi r^\star - \boldsymbol{Q}^\star\| \le (1 - \gamma^2)^{-1/2} \|\Pi \boldsymbol{Q}^\star - \boldsymbol{Q}^\star\|$.*

The theorem establishes the convergence of MARTA to Markov perfect equilibrium of $\mathcal{G}$ with the use of linear function approximators. The second statement bounds the proximity of the convergence point by the smallest approximation error that can be achieved given the choice of basis functions.

## 4   MARTA WITH A MALFUNCTION BUDGET

The fault-activation cost plays a critical role in ensuring that the Adversary induces malfunctions which cause a significant reduction in the system performance. Despite the intuitive interpretation, in

---

[3]A policy $(\pi^i, \pi^{-i}) = \boldsymbol{\pi} \in \boldsymbol{\Pi}$ is a Markov perfect equilibrium if no agent can improve the expected return by changing their current policy. Formally, $\forall i \in \mathcal{N}, \forall \pi'^i \in \Pi^i$, we have $v(s|\boldsymbol{\pi}, \mathfrak{g}) - v(s|(\pi'^i, \pi^{-i}), \mathfrak{g}) \le 0$.

some applications, the choice of $c$ is not obvious. In this section, we consider an alternative view and introduce a variant of MARTA, namely MARTA-B that imposes a budgetary constraint of the number of malfunctions that can be induced by Adversary during training. We show that by tracking its remaining budget the MARTA-B framework is able to learn a policy that makes optimal usage of its budget while respecting the budget constraint almost surely. The problem in which Switcher now faces a fixed budget on the number of malfunctions gives rise to the following constrained problem:

$$\min_{\hat{\mathfrak{g}}} \max_{\hat{\boldsymbol{\pi}} \in \boldsymbol{\Pi}} v_S(s|\hat{\boldsymbol{\pi}}, \hat{\mathfrak{g}}) \ \text{ s. t. } n - \sum_{k < \infty} \sum_{t_k \geq 0} \mathbf{1}(\mathfrak{g}(\cdot|s_{t_k})) \geq 0,$$

where $n \geq 0$ is a fixed value that represents the budget for the number of malfunctions and the index $k = 1, \ldots$ represents the training episode count. As in (Sootla et al., 2022; Mguni et al., 2023), we introduce a new variable $y_t$ that tracks the remaining number of activations: $y_t := n - \sum_{t \geq 0} \mathbf{1}(\mathfrak{g}(s_t))$ where the variable $y_t$ is now treated as the new state variable which is a component in an augmented state space $\mathcal{X} := \mathcal{S} \times \mathbb{N}$. We introduce the associated reward functions $\widetilde{\mathcal{R}} : \mathcal{X} \times \boldsymbol{\mathcal{A}} \to \mathcal{P}(D)$ and $\widetilde{\mathcal{R}}_S : \mathcal{X} \times \boldsymbol{\mathcal{A}} \times \mathcal{N} \times \{0\} \to \mathcal{P}(D)$ and the probability transition function $\widetilde{P} : \mathcal{X} \times \boldsymbol{\mathcal{A}} \times \mathcal{X} \to [0, 1]$ whose state space input is now replaced by $\mathcal{X}$ and Switcher value function for the game $\widetilde{\mathcal{G}} = \langle (\mathcal{N}, \text{Switcher}, \text{Adversary}), \mathcal{S}, ((\mathcal{A}_i)_{i \in \mathcal{N}}, \mathcal{A}_S, \mathcal{A}), \tilde{P}, (\widetilde{\mathcal{R}}, \widetilde{\mathcal{R}}_S, \widetilde{\mathcal{R}}_A), \gamma \rangle$. We now prove MARTA-B generates best-response FT policies for a given number of fault-activations (fault-activation budget).

**Theorem 5.** *Consider the budgeted problem* $\widetilde{\mathcal{G}}$, *then for any* $\widetilde{v} : \mathcal{X} \to \mathbb{R}$, *the solution of* $\widetilde{\mathcal{G}}$ *is given by*

$$\lim_{k \to \infty} \tilde{T}_G^k \widetilde{v}_S = \min_{\hat{\mathfrak{g}}} \max_{\hat{\boldsymbol{\pi}}} \widetilde{v}_S(\cdot|\hat{\boldsymbol{\pi}}, \hat{\mathfrak{g}}) = \max_{\hat{\boldsymbol{\pi}}} \min_{\hat{\mathfrak{g}}} \widetilde{v}_S(\cdot|\hat{\boldsymbol{\pi}}, \hat{\mathfrak{g}}),$$

*where Switcher's optimal policy takes the Markovian form* $\widetilde{\mathfrak{g}}(\cdot|\boldsymbol{x})$ *for any* $\boldsymbol{y} \equiv (y, s) \in \mathcal{X}$.

Theorem 5 proves, under standard assumptions, MARTA converges to the solution of Switcher's problem and the dec-POMDP under a Switcher fault-activation budget constraint.

## 5 Experiments

We conduct a series of experiments to verify: **(1) Plug-and-play effectiveness.** MARTA can be attached to standard MARL backbones without architectural changes and consistently improves robustness under injected malfunctions; **(2) Mechanism validity and calibration.** the learned *Switcher* meaningfully decides *when* and *whom* to intervene on, is tunable via the switch cost $c$. The effect is consistent across tasks, malfunction, different team sizes, and under train–test shifts.

**Environments. (1) Traffic Junction (TJ)** Koul (2019). A discrete grid-world in which agents must navigate through intersections without collisions. This environment emphasises high-density spatial conflict resolution and tests the agents' ability to handle localised, time-critical disruptions. **(2) Level-Based Foraging (LBF)** Christianos et al. (2020). Agents control units of a certain level, food items of varying levels are scattered across the map, and each agent aims to collect as much as possible. Crucially, an agent can only collect food if the cumulative level of the adjacent agents performing the "collect" action is greater than or equal to the food's level. It captures the dynamic spectrum between independent behaviour and tightly coupled cooperation under partial observability. **(3) Multi-Agent Particle Environment (MPE) – SimpleTag** Lowe et al. (2017). A continuous control domain where multiple pursuers must coordinate to capture the evader. This setting emphasises coordination under non-stationarity. Individual agents not coordinating effectively can severely reduce rewards.

**Experiment Configurations.** To make the robustness setting transparent, we factor our design into a few orthogonal axes. At a high level, each run is determined by: *who* malfunctions (agent selection), *how* the malfunction behaves (fault policy), the per-step trigger rate $p$, the switch cost $c$ that calibrates interventions, and whether the train–test fault processes are aligned or shifted. Table 1 summarises the factors and levels. At each timestep $t$, with probability $p$, one agent is selected to enter a malfunction state and execute a corrupted policy $\sigma^i$ for that step.

To facilitate analysis, we categorise our experiments into three difficulty bands. **Easy (Level 1)** settings correspond to aligned train–test distributions where a single fixed agent may malfunction with probability $p$. These include our core plug-and-play evaluations in TJ, LBF ($5 \times 5$–4p–1f), and MPE (SimpleTag). **Medium (Level 2)** settings also use aligned distributions but at each step the identity of the faultable agent is resampled, so the fault location varies across time while only one (or

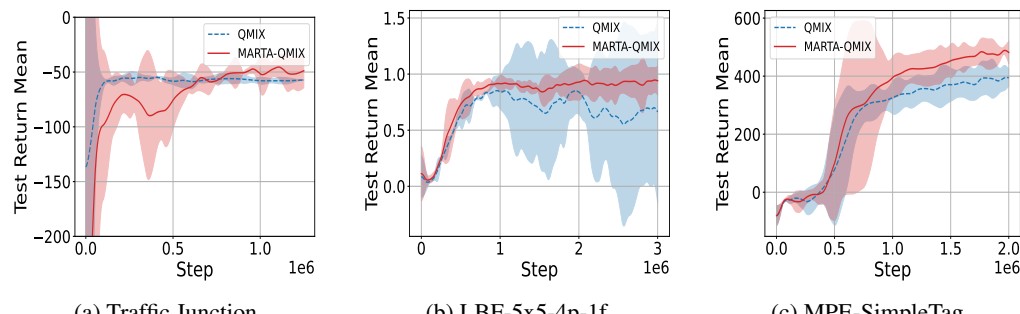

|                        |                 |                  |
|:----------------------:|:---------------:|:----------------:|
| (a) Traffic Junction   | (b) LBF-5x5-4p-1f | (c) MPE-SimpleTag |

Figure 2: **Robustness against malfunctions.** Each plot compares a base MARL algorithm with and without MARTA in faulty agent settings. In all scenarios, MARTA improves robustness.

less) agent may fail per step. Finally, **Hard (Level 3)** settings involve train–test shifts, either in the fault distribution, malfunction probability, or the fault policy. Here, each agent may malfunction, and the training and testing phases will have different malfunction distributions. Together these bands provide a structured view of robustness, ranging from predictable aligned malfunctions to adversarial distribution shifts with multiple concurrent faults.

**Evaluation Protocol.** Every $10,000$ steps, agents are evaluated for more than $100$ episodes using the same configuration of the malfunction. The best-performing checkpoint is reported based on average return. Unless otherwise specified, all hyperparameters and architecture configurations are identical across MARTA and its baseline counterpart. In all plots **dark lines represent averages over 3 seeds and the shaded regions represent** $95\%$ **confidence intervals**. We summarise where key experimental details are located:

- Environment configurations, agent counts and fault parameters are in Table 2 in the Appendix.
- Train–test fault protocols (aligned, shifted and resampled) are summarised in Table 1.
- Agent counts are as follows: TJ (4–6 agents), LBF (4 agents), and MPE SimpleTag (4 agents).
- Detailed descriptions of how faulty agents are generated and controlled are provided in Table 1.

CAN MARTA IMPROVE MARL ALGORITHM PERFORMANCE AND REDUCE FAILURE MODES?

To validate our claim that MARTA enhances robustness and improves performance, we evaluate MARTA on TJ, LBF (5x5-4p-1f), and MPE (SimpleTag) under injected malfunctions. In each environment, we compare a base MARL algorithm with its MARTA-augmented counterpart. As shown in Fig. 2, MARTA consistently improves robustness and return. In terms of final return, MARTA-QMIX achieves gains of **11.1%** in TJ, **21.4%** in MPE, and **44.6%** in LBF, with complete results reported in Table 3 in Sec. B. Robustness is particularly difficult in smaller-agent settings, where the failure of a single agent has a disproportionate impact on coordination. This effect is evident in TJ and LBF, where limited redundancy magnifies the influence of faults. Despite this, MARTA delivers consistent improvements, including in MPE, where continuous control and adversarial dynamics further increase task complexity. These results demonstrate MARTA enhances fault tolerance across diverse MARTA consistently improves fault tolerance across tasks while preserving the base learner's architecture and hyperparameters. In addition, **MARTA enables the baseline algorithm to avoid the dangers** of agent collisions in the TJ environment. In TJ we define the failure rate as the episode-level collision rate: $\text{TJ\_collision\_rate} = \frac{1}{M}\sum_{e=1}^{M}\mathbf{1}\{\text{episode } e \text{ contains any collision}\}$ where $M$ is the total number of episodes. We plotted the failure rates of each algorithm in Fig. 3(a), which shows that MARTA significantly reduces failure rates.

**The importance of Switching Controls.** A key component of MARTA is the switching control mechanism. This enables the Switcher to choose to activate malfunctions at high-risk states where learning robust behaviour is critical. To evaluate the impact of the switching control component, we compared the performance of MARTA with a variant of MARTA in which the Switcher's policy is replaced with a Bernoulli random trigger: for some $p \in (0, 1]$, at each state, a malfunction for agent $i \in \mathcal{N}$ occurs with probability $p/N$ (where $N = |\mathcal{N}|$), and with probability $1 - p$ no malfunction is activated. As shown in Fig. 3(b), incorporating the ability to learn an optimal switching control leads to much better performance compared to activating malfunctions randomly ("random policy").

**Experiment parameters.** In Sec. A, we also perform ablations on the switch cost $c$, which reveal that

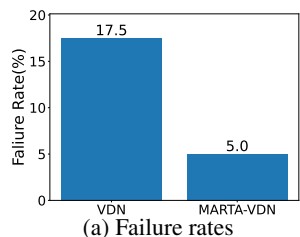 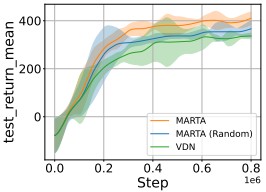

(a) Failure rates           (b) Effect of switcher policy

Figure 3: (a) Failure rates (collision rate per episode) of each algorithm in MPE. (b) Comparison of MARTA's switching control against random activations of malfunctions.

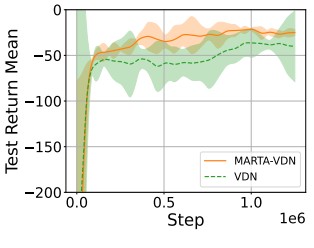 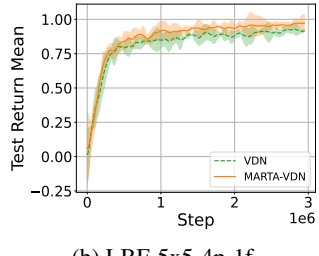 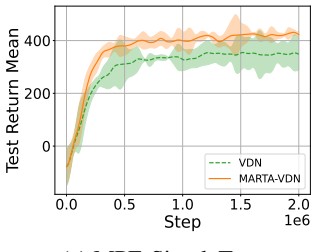

(a) Traffic Junction      (b) LBF-5x5-4p-1f      (c) MPE-SimpleTag

Figure 4: **Performance of MARTA+VDN.** MARTA improves performance in all scenarios.

adjusting this parameter increases the system's preference for FT in the fault–performance trade-off as $c \downarrow 0$. We further ablate the malfunction probability $p$ and the agent count $N$, demonstrating the robustness of MARTA to these experiment parameters.

**MARTA is a plug-and-play enhancement framework.** To validate our claim that MARTA easily adopts MARL algorithms, we tested the improvements MARTA delivers with an alternative base MARL algorithm. Specifically, we replaced QMIX in MARTA with VDN (Sunehag et al., 2017). Fig. 4 shows learning curves. In the TJ environment, MARTA-VDN yields a **37.8%** gain over its VDN baseline and in MPE, MARTA-VDN yields a **21.2%** gain over its VDN baseline. Indeed, we see that in all maps, MARTA-VDN significantly outperforms the base VDN algorithm.

**Comparison with robust adversarial MARL baselines.** We compared MARTA with other robust MARL methods. Specifically, we adapt M3DDPG (Li et al., 2019) as a robust baseline on the MPE SIMPLETAG environment and additionally construct a MARTA-enhanced variant, MARTA-MADDPG, by applying our switching framework to the MADDPG backbone. Figure 5 reports the learning curves of MADDPG, M3DDPG, and MARTA-MADDPG under the same malfunction process. Overall, these results support our claim that MARTA is compatible not only with purely cooperative learners (VDN, QMIX) but also with robust adversarial MARL algorithms: when applied on top of MADDPG, MARTA yields additional gains over an established robust baseline (M3DDPG) under the same threat model and environment setting. As Figure 6, in the LBF $10 \times 10$–4p–3f environment, we compared the performance of EIR and MARTA-VDN under two different fault scenarios. In Case 1, we followed the original EIR scenario: at most one agent may fail in each episode and remains unchanged within that episode. In this case, EIR and MARTA-VDN show similar final failure rates, but EIR converges faster. In Case 2, we created a training-test distribution inconsistency: during the testing phase, at each time step, each agent has a certain probability of being taken over by an adversary, meaning the location of the "potential fault" randomly switches between time and agent. As can be seen, under this more complex and dynamic fault mode, MARTA's failure rate is significantly lower than EIR, indicating that the MARTA framework has stronger robustness and generalization ability when facing dynamic, multi-location fault distributions.

**Computational overhead.** In all experiments, we reuse the same architectures and hyperparameters as the underlying MARL baselines and train the Switcher with a lightweight actor–critic learner. This introduces only a modest training-time overhead. At execution time there is no auxiliary safety filter or online optimisation so the wall-clock cost of MARTA is close to that of the base learner alone.

## 6 RELATED WORK

**Fault tolerance and safety in MARL.** Safety and robustness in MARL have been studied through shielding, backup policies and constrained optimisation. Shielding approaches (Zhang et al., 2019; ElSayed-Aly et al., 2021) use additional safety layers or backup policies to override unsafe actions.

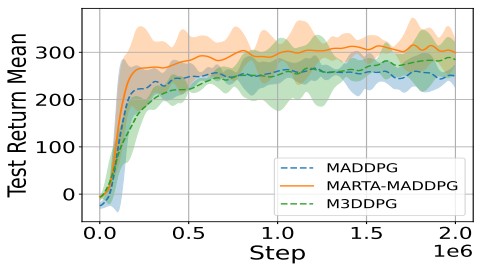

Figure 5: MPE.

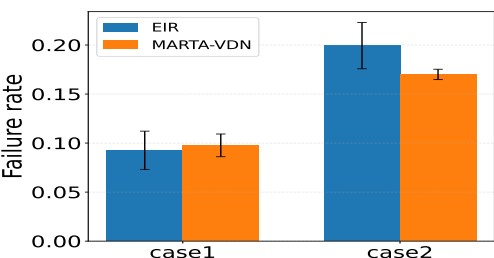

Figure 6: MARTA vs. EIR in LBF.

Qin et al. (Qin et al., 2021) employ control barrier functions to enforce safety constraints, but without formal guarantees in multi-agent settings. These methods often require per-timestep safety checks or dedicated certificates, and their cost grows with the number of agents. In contrast, MARTA embeds robustness directly into the training dynamics avoiding runtime safety layers and preserving the architecture of the underlying MARL learner. Constrained MARL formulations (Gu et al., 2021; Lu et al., 2021) treat safety as a constrained optimisation problem in an MG. These methods often face convergence and scalability challenges. By contrast, the game underlying MARTA has a unique solution to which MARTA has convergence guarantees for tabular and linearly approximated settings.

**Robust and adversarial MARL.** Adversarial training methods for RL and MARL (Pinto et al., 2017; Li et al., 2019; Zhang et al., 2020) typically introduce an opponent that perturbs actions, observations or dynamics to construct worst-case trajectories. These methods improve robustness but often induce overly conservative behaviour, since the agent is trained under an adversary that is active at every step. Moreover, most such work focuses on perturbations in a single-agent MDP or on model uncertainty, rather than on explicit agent malfunctions in cooperative teams. MARTA differs in three ways. First, it targets actuator-level failures in which an entire agent temporarily loses control to a fault policy, rather than small perturbations around nominal actions or states. Second, it models the timing and location of faults through a Switcher that explicitly reasons over state-dependent costs or budgets, rather than assuming an always-on adversary. Third, it provides convergence guarantees for this switching-augmented game, including under linear function approximation and budget constraints.

**Diagnostics and poisoning attacks in MARL.** Recent work has examined the vulnerability of MARL systems to targeted perturbations or poisoning attacks. RTCA (Zhou & Liu, 2023) proposes a resilience testing framework that perturbs the states of critical agents to expose weaknesses. Zheng et al. study training-time poisoning in which a single manipulated agent can poison policies. These works are primarily *diagnostic* or *attack-oriented*: they evaluate the weakness of existing MARL policies or design efficient attacks, rather than providing a defence scheme that yields robust policies. MARTA is complementary. It is a training-time defence mechanism using Switcher–Adversary pair to induce controlled, state-dependent malfunctions then trains agents to jointly best respond.

**Byzantine-robust MARL and adversarial teammates.** Li et al. (Li et al., 2023) study Byzantine-robust cooperative MARL through a Bayesian game formulation, in which some teammates may behave adversarially. Their focus is on strategic deviations modelled through adversarial types and on robust reasoning about such behaviour. MARTA instead models non-strategic actuator malfunctions, such as stuck actuators, corrupted control modules and state-dependent controller failures. These represent different robustness regimes. Strategic adversaries may deliberately coordinate to mislead, whereas actuator faults in physical systems are often non-strategic yet catastrophic for coordination. MARTA introduces a specific fault-switching MG (with budgets), and proves existence and uniqueness of a minimax value and convergence under both tabular and linearly approximated settings. This yields a different set of theoretical guarantees tailored to the malfunction setting.

**Shielding, backup policies and scalability.** Shielding and backup-policy methods (Zhang et al., 2019; ElSayed-Aly et al., 2021; Qin et al., 2021) provide valuable tools for enforcing safety constraints by modifying actions at execution time. However, their reliance on online constraint checking and per-agent safety mechanisms can create scalability challenges as the number of agents grows. MARTA takes a complementary approach which avoids runtime safety layers and allows robustness to scale with the underlying MARL learner without redesigning its architecture. Finally, MARTA is intentionally plug-and-play. It attaches to standard value-based and actor–critic MARL algorithms without altering their internal networks, and can also be combined with more sophisticated robust architectures. In this sense, MARTA acts as a general robustness layer that complements rather than replaces existing robust MARL techniques.

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

# Supplementary Material

## A  ABLATION STUDIES

**Switcher Effectiveness and Calibration.** We evaluate the effectiveness of MARTA's Switcher mechanism in two aspects: (1) Its ability to *calibrate* the frequency of adversarial malfunctions via the parameter $c$; (2) Its performance in Level 2 (training/test malfunction distribution shift):

**Varying Switching Cost $c$ (Fig. 7(a) and Fig. 7(c)).** We train MARTA under different switching costs $c$ in the TJ environment. Larger values of $c$ make Switcher more conservative in triggering malfunctions, resulting in fewer fault activations. This reflects a trade-off: high $c$ limits unnecessary disruptions but may reduce robustness; low $c$ encourages aggressive adversarial training. The monotonic trend validates that Switcher adaptively regulates the fault difficulty during training.

**Is MARTA robust under malfunction distributional shifts? (Fig. 7(b)).** We compare MARTA under **Case 1** (aligned malfunction distributions in training and testing) and **Case 2** (mismatched distributions), across different switching costs $c$ in the TJ environment. The performance gap between the cases highlights the generalisation difficulty under distributional shift.

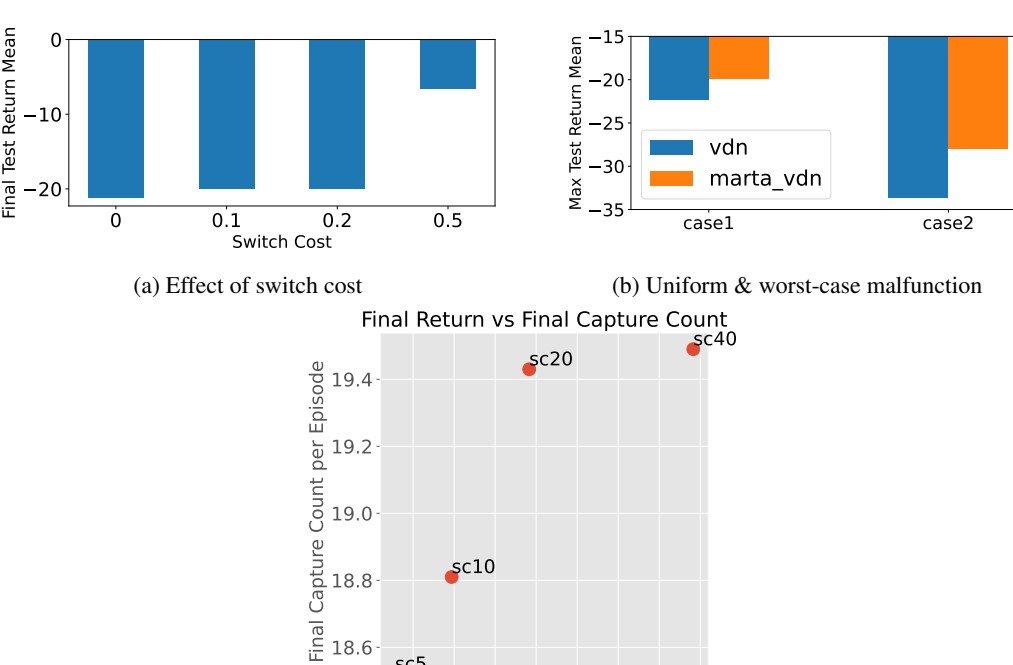

(a) Effect of switch cost          (b) Uniform & worst-case malfunction

(c) Trade-off between return and Collision rate

Figure 7: **Switcher ablation experiments.** (a) shows final evaluation returns under different switching costs $c$. (b) compares MARTA in Case 1 and Case 2.

**Is MARTA robust under Varying Malfunction Probability?** We evaluate the fault-tolerance of MARTA across different levels of malfunction probability $p$. Specifically, $p$ denotes the per-timestep probability that a malfunction is introduced i.e., at each timestep, with probability $p$, an agent becomes faulty and executes the malfunction policy. Larger $p$ values indicate more frequent and unpredictable failures, thereby increasing the difficulty of the task. We conduct this experiment in the **Traffic Junction** environment, using QMIX as the base learner. As shown in Fig. 8, we vary $p$ and compare the final returns of MARTA-QMIX with QMIX. The results show a clear performance gap: while the baseline QMIX degrades rapidly under higher malfunction frequencies, MARTA-QMIX does not suffer performance loss even at higher $p$. This suggests the Switcher mechanism in MARTA enables more effective adaptation to dynamic and persistent failure conditions. In summary, MARTA

significantly extends the robustness envelope of its underlying MARL algorithm, showing consistent advantage across a wide range of disturbance intensities.

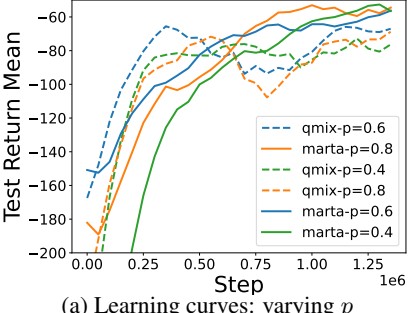

(a) Learning curves: varying $p$

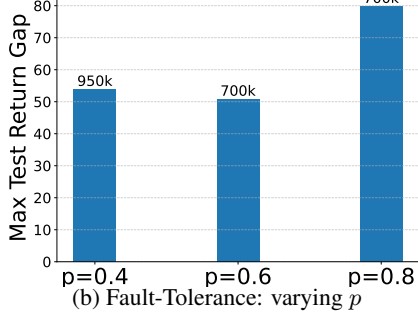

(b) Fault-Tolerance: varying $p$

Figure 8: Robustness under varying malfunction probability $p$ in the TJ environment.

# B FURTHER ANALYSIS OF EXPERIMENTAL RESULTS

Table 1: Experimental factors, levels, and rationale

| Factor | Levels | Rationale & Motivation |
|---|---|---|
| Agent selection (*who*) $F$ | {*Simple, Mid, Hard*} | **Simple**: same fixed agent $i^\star$ fails with per-step prob. $p$ in train/test; persistent, predictable single-point failure. **Mid**: per step, sample one agent $i_t \sim F$; train/test share $F$, unpredictable yet aligned. **Hard**: all agents faultable (concurrent allowed) and train/test fault processes mismatched ($F_{\text{test}} \neq F_{\text{train}}$, $p_{\text{test}} \neq p_{\text{train}}$); strongest generalisation stress. |
| Fault policy (*how*) $\sigma^i$ | {*Uniform, Worst-case*} | **Uniform**: uniformly random action; noise baseline. **Worst-case**: adversarial via $\text{softmax}(-Q)$; targets high-impact disruption without altering environment dynamics. |
| Trigger probability $p$ | grid $\mathcal{P}$ | Controls fault frequency (difficulty). Sweeps over $p \in \mathcal{P}$ quantify degradation and intervention–return trade-offs. |
| Switch cost $c$ | grid $\mathcal{C}$ | Calibrates Switcher activation; larger $c$ discourages frequent interventions. Sweeps test robustness–efficiency trade-off. |
| Train–test protocol | {*Aligned, Shifted*} | **Aligned (Case 1)**: test uses same $F, \sigma^i$ as training; ID robustness. **Shifted (Case 2)**: test alters $F$ and/or $\sigma^i$; OOD generalisation. |
| Switcher type (ablation) | {*Learned, Random*} | **Random**: fixed Bernoulli triggers with all else fixed; isolates the benefit of state-dependent, learned switching (Learned). |

Table 2: Settings for Fig. 2.

| Env | Base Algo (Agents) | Malfunction | $p$ | Fig |
|---|---|---|---|---|
| TJ | VDN (4) | Level2 | 0.1 | a |
| | QMIX (6) | Level1 | 0.1 | a |
| LBF | VDN (4) | Level2 | 0.4 | b |
| | QMIX (4) | Level2 | 0.2 | b |
| MPE | QMIX (4) | Level1 | 0.1 | c |
| | MADDPG (4) | Level2 | 0.4 | c |

To provide a quantitative summary, we evaluate each method using two metrics:

**Final Test Return:** We average the test return from the last 5 evaluation checkpoints for each seed, and report the mean ± standard deviation across seeds. As shown in Table 3, MARTA exhibits clear performance gains in all three scenarios. The column "Gain (%)" indicates the relative improvement of MARTA over its baseline, computed as the percentage change with respect to the baseline's absolute value, i.e., $(\text{MARTA} - \text{Baseline})/|\text{Baseline}| \times 100$.

**Area Under the Learning Curve (AUC):** To jointly evaluate sample efficiency and final return, we compute the AUC over the test return curve for each seed in Fig 9. MARTA consistently achieves higher AUC scores than its baselines, indicating both faster and more stable learning. The AUC "Gain (%)" reflects the relative increase in cumulative performance throughout training. Note that in TRAFFIC JUNCTION, MARTA-QMIX exhibits a lower AUC than QMIX because it converges more slowly in the early phase; however, it ultimately attains a higher final return.

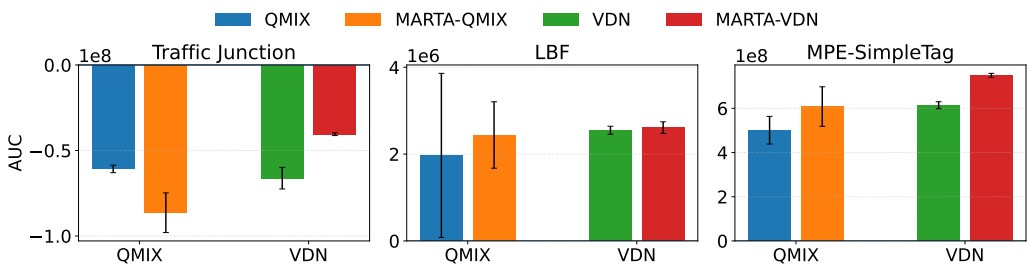

Figure 9: AUC comparisons across environments.

Table 3: The improvements of final return and AUC (± std) over baselines across 3 seeds.

| Env | Method | Final Return ($\mu \pm \sigma$) | Gain (%) | AUC ($\mu \pm \sigma$) | Gain (%) |
|---|---|---|---|---|---|
| Traffic Junction | QMIX | $-54.2 \pm 1.1$ | | $-6.08e7 \pm 2.19e6$ | |
| | MARTA-QMIX | $-48.2 \pm 4.7$ | +11.1 | $-8.64e7 \pm 11.6e6$ | -42.1 |
| | VDN | $-40.1 \pm 2.2$ | | $-6.62e7 \pm 6.31e6$ | |
| | MARTA-VDN | $-24.9 \pm 0.8$ | +37.8 | $-4.05e7 \pm 0.79e6$ | +38.8 |
| LBF | QMIX | $0.65 \pm 0.279$ | | $1.97e6 \pm 18.9e5$ | |
| | MARTA-QMIX | $0.94 \pm 0.043$ | +44.6 | $2.44e6 \pm 7.65e5$ | +23.9 |
| | VDN | $0.94 \pm 0.002$ | | $2.55e6 \pm 0.92e5$ | |
| | MARTA-VDN | $0.97 \pm 0.021$ | +3.2 | $2.61e6 \pm 1.33e5$ | +2.4 |
| MPE-SimpleTag | QMIX | $395.28 \pm 14.78$ | | $5.01e8 \pm 6.26e7$ | |
| | MARTA-QMIX | $479.79 \pm 26.19$ | +21.4 | $6.08e8 \pm 8.94e7$ | +21.4 |
| | VDN | $346.80 \pm 19.01$ | | $6.14e8 \pm 1.58e7$ | |
| | MARTA-VDN | $420.26 \pm 5.789$ | +21.2 | $7.49e8 \pm 0.92e7$ | +22.0 |

# C    ALGORITHMS

In this section, we provide the pseudocode for 3 variants of MARTA, namely an actor-critic variant of MARTA called MARTA-AC (Algorithm 1) a Q learning variant called MARTA-Q (Algorithm 2) and, lastly a version of MARTA-AC that accommodates budget constraints called MARTA-Budget (Algorithm 3).

---

**Algorithm 1: M**ulti-**A**gent **R**obust **T**raining **A**lgorithm (MARTA)-AC

---
1: **Input:** Stepsize $\alpha$, batch size $B$, episodes $K$, steps per episode $T$, mini-epochs $e$, malfunction cost $c$, Termination probability parameter $p$ (Bernoulli distributed).
2: **Initialise:** Policy network (acting) $\boldsymbol{\pi}$, Policy network (switching) $\mathfrak{g}$,
   Policy network (adversary) $(\sigma^1, \ldots, \sigma^N) = \boldsymbol{\sigma}$, Critic network (acting )$V_{\boldsymbol{\pi}}$, Critic network (switching )$V_{\mathfrak{g}}$, Critic network (adversary )$V_{\boldsymbol{\sigma}}$, for any $t < 0$ set termination probability $p_t \equiv 1$.
3: Given reward objective function, $\mathcal{R}$, initialise Rollout Buffers $\mathcal{B}_{\pi}$, $\mathcal{B}_{\mathfrak{g}}$ (use Replay Buffer for SAC), $\mathcal{B}_{\boldsymbol{\sigma}}$.
4: **for** $N_{episodes}$ **do**
5:    Reset state $s_0$, Reset Rollout Buffers $\mathcal{B}_{\pi}$, $\mathcal{B}_{\mathfrak{g}}$, $\mathcal{B}_{\boldsymbol{\sigma}}$
6:    **for** $t = 0, 1, \ldots$ **do**
7:       Sample $(f_t^1, \ldots, f_t^N) = \boldsymbol{f}_t \sim \boldsymbol{\sigma}(\cdot|s_t)$, $\boldsymbol{a}_t \sim \boldsymbol{\pi}(\cdot|s_t)$, $g_t \sim \mathfrak{g}(\cdot|s_t)$, $p_t \sim \text{Bern}(p)$
8:       **if** $(1 - p_{t-1})g_{t-1} > 0$ (hence $(1 - p_{t-1})g_{t-1} = i \in \mathcal{N}$) **then**
9:          Set $g_t \equiv g_{t-1}$. Apply $f_t^i$ and $a_t^{-i}$ where $i \equiv g_{t-1}$ so $s_{t+1} \sim P(\cdot|(f_t^i, a_t^{-i}), s_t)$,
10:         Receive rewards $\boldsymbol{r}_{S,t} = -c - \boldsymbol{r}_t$ and $\boldsymbol{r}_t$ where $\boldsymbol{r}_t \sim \mathcal{R}(s_t, (f_t^i, a_t^{-i}))$
11:         Store $(s_t, (f_t^i, a_t^{-i}), s_{t+1}, \boldsymbol{r}_t)$, $(s_t, (f_t^i, a_t^{-i}), s_{t+1}, -\boldsymbol{r}_t)$ and $(s_t, g_t, s_{t+1}, \boldsymbol{r}_{S,t})$ in $\mathcal{B}_{\boldsymbol{\pi}}, \mathcal{B}_{\boldsymbol{\sigma}}$ and $\mathcal{B}_{\mathfrak{g}}$ respectively.
12:      **else**
13:         **if** $g_t = i \in \{1, \ldots, N\}$ **then**
14:            Apply $f_t^i$ and $a_t^{-i}$ so $s_{t+1} \sim P(\cdot|(f_t^i, a_t^{-i}), s_t)$,
15:            Receive rewards $\boldsymbol{r}_{S,t} = -c - \boldsymbol{r}_t$ and $\boldsymbol{r}_t$ where $\boldsymbol{r}_t \sim \mathcal{R}(s_t, (f_t^i, a_t^{-i}))$
16:            Store $(s_t, (f_t^i, a_t^{-i}), s_{t+1}, \boldsymbol{r}_t)$, $(s_t, (f_t^i, a_t^{-i}), s_{t+1}, -\boldsymbol{r}_t)$ and $(s_t, g_t, s_{t+1}, \boldsymbol{r}_{S,t})$ in $\mathcal{B}_{\boldsymbol{\pi}}, \mathcal{B}_{\boldsymbol{\sigma}}$ and $\mathcal{B}_{\mathfrak{g}}$ respectively.
17:         **else**
18:            Apply the actions $\boldsymbol{a}_t \sim \boldsymbol{\pi}(\cdot|s_t)$ so $s_{t+1} \sim P(\cdot|\boldsymbol{a}_t, s_t)$,
19:            Receive rewards $\boldsymbol{r}_{S,t} = -\boldsymbol{r}_t$ and $\boldsymbol{r}_t$ where $\boldsymbol{r}_t \sim \mathcal{R}(s_t, \boldsymbol{a}_t)$.
20:         **end if**
21:      **end if**
22:      Store $(s_t, \boldsymbol{a}_t, s_{t+1}, \boldsymbol{r}_t)$ and $(s_t, g_t, s_{t+1}, \boldsymbol{r}_{S,t})$ in $\mathcal{B}_{\boldsymbol{\pi}}$ and $\mathcal{B}_{\mathfrak{g}}$ respectively.
23:   **end for**
24:   **// Learn the individual policies**
25:   Update policy $\boldsymbol{\pi}$ and critic $V_{\boldsymbol{\pi}}$ networks using $\mathfrak{B}_{\boldsymbol{\pi}}$
26:   Update policy $\mathfrak{g}$ and critic $V_{\mathfrak{g}}$ networks using $\mathfrak{B}_{\mathfrak{g}}$
27:   Update policy $\boldsymbol{\sigma}$ and critic $V_{\boldsymbol{\sigma}}$ networks using $\mathfrak{B}_{\boldsymbol{\sigma}}$
28: **end for**

---

**Algorithm 2: M**ulti-**A**gent **R**obust **T**raining **A**lgorithm (MARTA)- Q

1: **Input:** Constant $\epsilon \geq 0$,
2: **Initialise:** Q-function, $\boldsymbol{Q}_0$, set termination probability $p_t \equiv 1$ for any $t < 0$
3: $n \leftarrow 0$
4: **for** $t = 0, 1, \ldots$ **do**
5:     Estimate $\boldsymbol{a}_t \in \arg\max \boldsymbol{Q}_n(s_t, \boldsymbol{a}_t)$
6:     Estimate $f_t^i \in \underset{i \in \mathcal{N}, a_t^i \in \mathcal{A}^i}{\arg\min} \; \underset{a_t^{-i} \in \mathcal{A}^{-i}}{\max} Q_S(s, (a_t^i, a_t^{-i}), g)$
7:     **if** $(1 - p_{t-1})g_{t-1} > 0$ (i.e. $(1 - p_{t-1})g_{t-1} \in \mathcal{N}$) **then**
8:         Set $g_t \equiv g_{t-1}$. Apply $f_t^i$ and $a_t^{-i}$ where $i \equiv g_{t-1}$ so $s_{t+1} \sim P(\cdot | (f_t^i, a_t^{-i}), s_t)$,
9:     **else**
10:         **if** $\hat{\mathcal{M}}\boldsymbol{Q}_n \geq \boldsymbol{Q}_n$ **then**
11:             Apply $f_t^i$ and $a_t^{-i}$ so $s_{t+1} \sim P(\cdot | (f_t^i, a_t^{-i}), s_t)$
12:             Receive rewards $\boldsymbol{r}_{S,t} = -c - \boldsymbol{r}_t$ and $\boldsymbol{r}_t$ where $\boldsymbol{r}_t \sim \mathcal{R}(s_t, (f_t^i, a_t^{-i}))$
13:         **else**
14:             Apply the actions $\boldsymbol{a}_t$ so $s_{t+1} \sim P(\cdot | \boldsymbol{a}_t, s_t)$,
15:             Receive rewards $\boldsymbol{r}_{S,t} = -\boldsymbol{r}_t$ and $\boldsymbol{r}_t$ where $\boldsymbol{r}_t \sim \mathcal{R}(s_t, \boldsymbol{a}_t)$.
16:         **end if**
17:     **end if**
18: **end for**
19: **// Learn** $\hat{Q}$
20: Update $\boldsymbol{Q}_n$ function according to the update rule equation 2

---

**Algorithm 3: M**ulti-**A**gent **R**obust **T**raining **A**lgorithm (MARTA)-Budget

1: **Input:** Stepsize $\alpha$, batch size $B$, episodes $K$, steps per episode $T$, mini-epochs $e$, malfunction cost $c$.
2: **Initialise:** Policy network (acting) $\boldsymbol{\pi}$, Policy network (switching) $\mathfrak{g}$, Policy network (adversary) $(\sigma^1, \ldots, \sigma^N) = \boldsymbol{\sigma}$, Critic network (acting )$V_{\boldsymbol{\pi}}$, Critic network (switching )$V_{\mathfrak{g}}$, Critic network (adversary )$V_{\boldsymbol{\sigma}}$.
3: Given reward objective function, $\mathcal{R}$, initialise Rollout Buffers $\mathcal{B}_{\boldsymbol{\pi}}$, $\mathcal{B}_{\mathfrak{g}}$ (use Replay Buffer for SAC), $\mathcal{B}_{\boldsymbol{\sigma}}$.
4: **for** $N_{episodes}$ **do**
5:     Reset state $s_0$, Reset Rollout Buffers $\mathcal{B}_{\boldsymbol{\pi}}$, $\mathcal{B}_{\mathfrak{g}}$, $\mathcal{B}_{\boldsymbol{\sigma}}$
6:     **for** $t = 0, 1, \ldots$ **do**
7:         Sample $(f_t^1, \ldots, f_t^N) = \boldsymbol{f}_t \sim \boldsymbol{\sigma}(\cdot | \boldsymbol{s}_t)$, $\boldsymbol{a}_t \sim \boldsymbol{\pi}(\cdot | \boldsymbol{s}_t)$, $g_t \sim \mathfrak{g}(\cdot | \boldsymbol{s}_t, \boldsymbol{f}_t, \boldsymbol{a}_t)$ $\boldsymbol{s}_t = (s_t, y_t)$
8:         **if** $g_t = i \in \{1, \ldots, N\}$ **then**
9:             Apply $f_t^i$ and $a_t^{-i}$ so $s_{t+1} \sim \widetilde{P}(\cdot | (f_t^i, a_t^{-i}), \boldsymbol{s}_t)$
10:            Receive rewards $\widetilde{\boldsymbol{r}}_{S,t} = -c - \boldsymbol{r}_t$ and $\widetilde{\boldsymbol{r}}_t$ where $\widetilde{\boldsymbol{r}}_t \sim \widetilde{\mathcal{R}}(\boldsymbol{s}_t, (f_t^i, a_t^{-i}))$,
11:            Store $(\boldsymbol{z}_t, f_t^i, a_t^{-i}, \boldsymbol{s}_{t+1}, \boldsymbol{r}_{S,t}, \boldsymbol{r}_t)$ in $\mathcal{B}_{\boldsymbol{\pi}}, \mathcal{B}_{\boldsymbol{\sigma}}, \mathcal{B}_{\mathfrak{g}}$ where $\boldsymbol{z}_t := (\boldsymbol{s}_t, g_t)$, $\boldsymbol{s}_t = (s_t, y_t)$.
12:         **else**
13:            Apply the actions $\boldsymbol{a}_t \sim \boldsymbol{\pi}(\cdot | \boldsymbol{s}_t)$ so $s_{t+1} \sim P(\cdot | \boldsymbol{a}_t, \boldsymbol{s}_t)$,
14:            Receive rewards $\widetilde{\boldsymbol{r}}_{S,t} = -\widetilde{\boldsymbol{r}}_t$ and $\widetilde{\boldsymbol{r}}_t$ where $\widetilde{\boldsymbol{r}}_t \sim \widetilde{\mathcal{R}}(s_t, \boldsymbol{a}_t)$.
15:         **end if**
16:         Store $(\boldsymbol{z}_t, \boldsymbol{a}_t, \boldsymbol{s}_{t+1}, \boldsymbol{r}_{S,t}, \boldsymbol{r}_t)$ in $\mathcal{B}_{\boldsymbol{\pi}}, \mathcal{B}_{\boldsymbol{\sigma}}, \mathcal{B}_{\mathfrak{g}}$.
17:     **end for**
18:     **// Learn the individual policies**
19:     Update policy $\boldsymbol{\pi}$ and critic $V_{\boldsymbol{\pi}}$ networks using $\mathfrak{B}_{\boldsymbol{\pi}}$
20:     Update policy $\mathfrak{g}$ and critic $V_{\mathfrak{g}}$ networks using $\mathfrak{B}_{\mathfrak{g}}$
21:     Update policy $\boldsymbol{\sigma}$ and critic $V_{\boldsymbol{\sigma}}$ networks using $\mathfrak{B}_{\boldsymbol{\sigma}}$
22: **end for**

## C.1 Computational Requirements

All experiments presented in this work were executed purely on CPUs. The experiments were executed in compute clusters that consist of several nodes. The main types of CPU models that were used for this work are GHz Quad-Core Intel Core i5 processor, Intel Iris Plus graphics. All experiments were executed using a single CPU core. The total number of CPU hours that were spent for executing the experiments in this work (excluding the hyper-parameter search) are 80,900.

## C.2 Hyperparameter Settings

In the table below we report all hyperparameters used in our experiments. Hyperparameter values in square brackets indicate ranges of values that were used for performance tuning.

| | |
|---|---|
| Clip Gradient Norm | 1 |
| $\gamma_E$ | 0.99 |
| $\lambda$ | 0.95 |
| Learning rate | $1 \times 10^{-4}$ |
| Number of minibatches | 4 |
| Number of optimisation epochs | 4 |
| Number of parallel actors | 16 |
| Optimisation algorithm | Adam |
| Rollout length | 128 |
| Sticky action probability | 0.25 |
| Use Generalised Advantage Estimation | True |
| Coefficient of extrinsic reward | [1, 5] |
| Coefficient of intrinsic reward | [1, 2, 5, 10, 20, 50] |
| Switcher discount factor | 0.99 |
| Probability of terminating option | [0.5, 0.75, 0.8, 0.9, 0.95] |
| $L$ function output size | [2, 4, 8, 16, 32, 64, 128, 256] |

# D PROOF OF TECHNICAL RESULTS

## D.1 NOTATION & ASSUMPTIONS

We assume that $\mathcal{S}$ is defined on a probability space $(\Omega, \mathcal{F}, \mathbb{P})$ and any $s \in \mathcal{S}$ is measurable with respect to the Borel $\sigma$-algebra associated with $\mathbb{R}^p$. We denote the $\sigma$-algebra of events generated by $\{s_t\}_{t \geq 0}$ by $\mathcal{F}_t \subset \mathcal{F}$. In what follows, we denote by $(\mathcal{Y}, \|\|)$ any finite normed vector space and by $\mathcal{H}$ the set of all measurable functions. The results of the paper are built under the following assumptions which are standard within RL and stochastic approximation methods:

**Assumption 1.** The stochastic process governing the system dynamics is ergodic, that is the process is stationary and every invariant random variable of $\{s_t\}_{t \geq 0}$ is equal to a constant with probability $1$.

**Assumption 2**. The agents' reward function $\mathcal{R}$ is in $L_2$.

**Assumption 3.** For any Switcher policy $\mathfrak{g}$, the total number of interventions is $K < \infty$.

We begin the analysis with some preliminary results and definitions required for proving our main results.

**Definition 1.** *A.1 Given a norm $\| \cdot \|$, an operator $T : \mathcal{Y} \to \mathcal{Y}$ is a contraction if there exists some constant $c \in [0, 1[$ for which for any $J_1, J_2 \in \mathcal{Y}$ the following bound holds: $\|TJ_1 - TJ_2\| \leq c\|J_1 - J_2\|$.*

**Definition 2.** *A.2 An operator $T : \mathcal{Y} \to \mathcal{Y}$ is non-expansive if $\forall J_1, J_2 \in \mathcal{Y}$ the following bound holds: $\|TJ_1 - TJ_2\| \leq \|J_1 - J_2\|$.*

**Lemma 1.** *(Mguni, 2019) For any $f : \mathcal{Y} \to \mathbb{R} : \mathcal{Y} \to \mathbb{R}$, we have that the following inequality holds:*

$$\left\| \max_{a \in \mathcal{Y}} f(a) - \max_{a \in \mathcal{Y}} g(a) \right\| \leq \max_{a \in \mathcal{Y}} \|f(a) - g(a)\|. \tag{2}$$

**Lemma 2.** *A.4(Tsitsiklis & Van Roy, 1999) The probability transition kernel $P$ is non-expansive so that if $\forall J_1, J_2 \in \mathcal{Y}$ the following holds: $\|PJ_1 - PJ_2\| \leq \|J_1 - J_2\|$.*

**Lemma 3.** *A.1 Define $\mathrm{val}^+[f] := \min_{b \in \mathbb{B}} \max_{a \in \mathbb{A}} f(a, b)$ and define $\mathrm{val}^-[f] := \max_{a \in \mathbb{A}} \min_{b \in \mathbb{B}} f(a, b)$, then for any $b \in \mathbb{B}$ we have that for any $f, g \in \mathbb{L}$ and for any $k \in \mathbb{R}_{>0}$:*

$$\left| \max_{a \in \mathbb{A}} f(a, b) - \max_{a \in \mathbb{A}} g(a, b) \right| \leq k \implies \left| \mathrm{val}^-[f] - \mathrm{val}^-[g] \right| \leq k.$$

**Lemma 4.** *A.2 For any $f, g, h \in \mathbb{L}$ and for any $k \in \mathbb{R}_{>0}$ we have that:*

$$\|f - g\| \leq k \implies \|\min\{f, h\} - \min\{g, h\}\| \leq k.$$

**Lemma 5.** *A.3 Let the functions $f, g, h \in \mathbb{L}$ then*

$$\|\max\{f, h\} - \max\{g, h\}\| \leq \|f - g\|. \tag{3}$$

*Proof of Lemma 3.* We begin by noting the following inequality for any $f : \mathcal{V} \times \mathcal{V} \to \mathbb{R}, g : \mathcal{V} \times \mathcal{V} \to \mathbb{R}$ s.th. $f, g \in \mathbb{L}$ we have by Lemma 4, that for all $b \in \mathcal{V}$:

$$\left| \max_{a \in \mathcal{V}} f(a, b) - \max_{a \in \mathcal{V}} g(a, b) \right| \leq \max_{a \in \mathcal{V}} |f(a, b) - g(a, b)|. \tag{4}$$

From (4) we can straightforwardly derive the fact that for any $b \in \mathcal{V}$:

$$\left| \min_{a \in \mathcal{V}} f(a, b) - \min_{a \in \mathcal{V}} g(a, b) \right| \leq \max_{a \in \mathcal{V}} |f(a, b) - g(a, b)|, \tag{5}$$

(this can be seen by negating each of the functions in (4) and using the properties of the $\max$ operator).

Assume that for any $b \in \mathcal{V}$ the following inequality holds:

$$\max_{a \in \mathcal{V}} |f(a, b) - g(a, b)| \leq k \tag{6}$$

Since (5) holds for any $b \in \mathcal{V}$ and, by (4), we have in particular that

$$\left| \max_{b \in \mathcal{V}} \min_{a \in \mathcal{V}} f(a,b) - \max_{b \in \mathcal{V}} \min_{a \in \mathcal{V}} g(a,b) \right|$$

$$\leq \max_{b \in \mathcal{V}} \left| \min_{a \in \mathcal{V}} f(a,b) - \min_{a \in \mathcal{V}} g(a,b) \right|$$

$$\leq \max_{b \in \mathcal{V}} \max_{a \in \mathcal{V}} |f(a,b) - g(a,b)| \leq k, \tag{7}$$

whenever (6) holds which gives the required result. $\qquad\square$

The following result fully characterises Switcher's policy $\mathfrak{g}$ in terms of an obstacle condition which can be determined at each state:

**Proposition 2.** *Denote by* $\boldsymbol{a} \equiv (a^i, a^{-i}) \in \boldsymbol{\mathcal{A}}$, *the optimal Switcher policy* $\mathfrak{g}^*$ *is given by* $\mathfrak{g}^*(\cdot|s) =$
$\arg\min_{i \in \mathcal{N}} \left[ \min_{a^i \in \mathcal{A}^i} \max_{a^{-i} \in \mathcal{A}^{-i}} Q_S(s, \boldsymbol{a}, g|\cdot) \right] \mathbf{1} \left( \max_{\boldsymbol{a} \in \boldsymbol{\mathcal{A}}} Q_S(s, \boldsymbol{a}, g|\cdot) - \hat{\boldsymbol{\mathcal{M}}} Q_S(s, \boldsymbol{a}, g|\cdot) \right)$ *for any* $s \in \mathcal{S}$
*where* $\mathbf{1}_{\mathbb{R}_+}(x) = 1$ *if* $x > 0$ *and* $0$ *otherwise.*

Prop. 2 provides characterisation of where Switcher should activate Adversary and which agent $i \in \mathcal{N}$ should be activated. The condition allows for the characterisation to be evaluated online during the learning phase.

## PROOF OF THEOREM 2

*Proof.* We begin by recalling the definition of the intervention operator $\hat{\boldsymbol{\mathcal{M}}}$ for any $s \in \mathcal{S}$:

$$\hat{\boldsymbol{\mathcal{M}}} Q_S(s, \boldsymbol{a}, g|\cdot) := c + \min_{i \in \mathcal{N}, a^i \in \mathcal{A}^i} \max_{a^{-i} \in \mathcal{A}^{-i}} Q_S(s, (a^i, a^{-i}), g|\cdot) \tag{8}$$

Secondly, recall that the Bellman operator for the game $\mathcal{G}$ acting on a function $v_S : \mathcal{S} \to \mathbb{R}$ is:

$$T v_S(s) := \min \left\{ \hat{\boldsymbol{\mathcal{M}}} Q_S, \max_{\boldsymbol{a} \in \boldsymbol{\mathcal{A}}} \left[ \mathcal{R}_S + \gamma \sum_{s' \in \mathcal{S}} P(s'; \cdot) v_S(s') \right] \right\} \tag{9}$$

It suffices to prove that $T$ is a contraction operator. Thereafter, we use both results to prove the existence of a fixed point for $\mathcal{G}$ as a limit point of a sequence generated by successively applying the Bellman operator to a test value function. Therefore our next result shows that the following bounds holds:

**Lemma 6.** *The Bellman operator* $T$ *is a contraction so that for any real-valued maps* $v_S, v_S'$, *the following bound holds:* $\|T v_S - T v_S'\| \leq \gamma \|v_S - v_S'\|$.

In the following proofs, we use the shorthand notation: $\mathcal{P}_{ss'}^{\boldsymbol{a}} =: \sum_{s' \in \mathcal{S}} P(s'; \boldsymbol{a}, s)$ and $\mathcal{P}_{ss'}^{\boldsymbol{\pi}} =: \sum_{\boldsymbol{a} \in \boldsymbol{\mathcal{A}}} \boldsymbol{\pi}(\boldsymbol{a}|s) \mathcal{P}_{ss'}^{\boldsymbol{a}}$. To prove that $T$ is a contraction, we consider the three cases produced by equation 9, that is to say we prove the following statements:

i) $\left| \max_{\boldsymbol{a} \in \boldsymbol{\mathcal{A}}} \left( \mathcal{R}_S(s_t, \boldsymbol{a}, g) + \gamma \mathcal{P}_{s' s_t}^{\boldsymbol{a}} v_S(s') \right) - \max_{\boldsymbol{a} \in \boldsymbol{\mathcal{A}}} \left( \mathcal{R}_S(s_t, \boldsymbol{a}, g) + \gamma \mathcal{P}_{s' s_t}^{\boldsymbol{a}} v_S'(s') \right) \right| \leq$
$\gamma \|v_S - v_S'\|$

ii) $\left\| \hat{\boldsymbol{\mathcal{M}}} Q_S - \hat{\boldsymbol{\mathcal{M}}} Q_S' \right\| \leq \gamma \|v_S - v_S'\|$, .

iii) $\left\| \hat{\boldsymbol{\mathcal{M}}} Q_S - \max_{\boldsymbol{a} \in \boldsymbol{\mathcal{A}}} \left[ \mathcal{R}_S(s_t, \boldsymbol{a}, g) + \gamma \mathcal{P}_{s's}^{\boldsymbol{a}} v_S' \right] \right\| \leq \gamma \|v_S - v_S'\|$.

We begin by proving i).

Indeed, for any $\boldsymbol{a} \in \mathcal{A}$ and $\forall s_t \in \mathcal{S}, \forall s' \in \mathcal{S}$ we have that

$$\left| \max_{\boldsymbol{a} \in \mathcal{A}} \left( \mathcal{R}_S(s_t, \boldsymbol{a}, g) + \gamma \mathcal{P}^{\pi}_{s' s_t} v_S(s') \right) - \max_{\boldsymbol{a} \in \mathcal{A}} \left( \mathcal{R}_S(s_t, \boldsymbol{a}, g) + \gamma \mathcal{P}^{\boldsymbol{a}}_{s' s_t} v'_S(s') \right) \right|$$

$$\leq \max_{\boldsymbol{a} \in \mathcal{A}} \left| \gamma \mathcal{P}^{\boldsymbol{a}}_{s' s_t} v_S(s') - \gamma \mathcal{P}^{\boldsymbol{a}}_{s' s_t} v'_S(s') \right|$$

$$\leq \gamma \left\| P v_S - P v'_S \right\|$$

$$\leq \gamma \left\| v_S - v'_S \right\|,$$

using the non-expansiveness of the operator $P$ and Lemma 1.

We now prove ii). Using the definition of $\boldsymbol{\mathcal{M}}$ we have that for any $s \in \mathcal{S}$

$$\left| (\hat{\boldsymbol{\mathcal{M}}} Q_S - \hat{\boldsymbol{\mathcal{M}}} Q'_S)(s, \boldsymbol{a}) \right|$$

$$= \left| \min_{i \in \mathcal{N}, a^i \in \mathcal{A}^i} \max_{a^{-i} \in \mathcal{A}^{-i}} \left( \mathcal{R}_S(s, \boldsymbol{a}, g) + c + \gamma \mathcal{P}^{\boldsymbol{a}}_{s' s} v_S(s) \right) - \min_{i \in \mathcal{N}, a^i \in \mathcal{A}^i} \max_{a^{-i} \in \mathcal{A}^{-i}} \left( \mathcal{R}_S(s, \boldsymbol{a}, g) + c + \gamma \mathcal{P}^{\boldsymbol{a}}_{s' s} v'_S(s) \right) \right|$$

$$\leq \gamma \max_{i \in \mathcal{N}, a^i \in \mathcal{A}^i} \max_{a^{-i} \in \mathcal{A}^{-i}} \left| \gamma \mathcal{P}^{a^i} \mathcal{P}^{a^{-i}} v_S(s) - \mathcal{P}^{a^i} \mathcal{P}^{a^{-i}} v'_S(s) \right|$$

$$\leq \gamma \max_{\boldsymbol{a} \in \mathcal{A}} \left| \mathcal{P}^{a^i} \mathcal{P}^{a^{-i}} v_S(s) - \mathcal{P}^{a^i} \mathcal{P}^{a^{-i}} v'_S(s) \right|$$

$$\leq \gamma \left\| P v_S - P v'_S \right\|$$

$$\leq \gamma \left\| v_S - v'_S \right\|,$$

using equation 5 in the second step and the fact that $P$ is non-expansive.

We now prove iii). In the proof of iii), we use the following equivalent representation of Switcher objective which is to find $(\hat{\boldsymbol{\pi}}, \hat{\mathfrak{g}})$ such that

$$(\hat{\boldsymbol{\pi}}, \hat{\mathfrak{g}}) \in \underset{\boldsymbol{\pi}, \mathfrak{g}}{\arg \max} \ \mathfrak{v}_S(s | \boldsymbol{\pi}, \mathfrak{g})$$

$$\mathfrak{v}_S(s | \boldsymbol{\pi}, \mathfrak{g}) = \mathbb{E}_{g \sim \mathfrak{g}} \left[ \sum_{t=0}^{\infty} \gamma^t \left( -\mathcal{R}(s_t, \boldsymbol{a}_t) - c \cdot \mathbf{1}_{\mathcal{N}}(g(s_t)) \right) \Big| s_0 = s; \boldsymbol{a} \sim \boldsymbol{\pi} \right].$$

Now the intervention operator transforms as $\hat{\boldsymbol{M}}$:

$$\hat{\boldsymbol{M}} Q_S(s, \boldsymbol{a}, g | \cdot) := -c + \max_{i \in \mathcal{N}, a^i \in \mathcal{A}^i} \min_{a^{-i} \in \mathcal{A}^{-i}} Q_S(s, (a^i, a^{-i}), g | \cdot), \tag{10}$$

for any $s \in \mathcal{S}$ and any $\boldsymbol{a} \in \mathcal{A}$. Similarly, the Bellman operator acting on a function $v_S : \mathcal{S} \to \mathbb{R}$ becomes:

$$\mathcal{T} v_S(s) := \min \left\{ \hat{\boldsymbol{M}} Q_S, \min_{\boldsymbol{a} \in \mathcal{A}} \left[ -\mathcal{R}_S + \gamma \sum_{s' \in \mathcal{S}} P(s'; \cdot) v_S(s') \right] \right\}, \forall s \in \mathcal{S}. \tag{11}$$

Therefore, to prove (iii), we must show that:

$$\left\| \hat{\boldsymbol{M}} Q_S - \min_{\boldsymbol{a} \in \mathcal{A}} \left[ -\mathcal{R}_S(s, \boldsymbol{a}, g) + \gamma \mathcal{P}^{\boldsymbol{a}}_{s' s} v'_S \right] \right\| \leq \gamma \left\| v_S - v'_S \right\|. \tag{12}$$

We split the proof of the statement into two cases:

**Case 1:** First, assume that for any $s \in \mathcal{S}$ and $\forall \boldsymbol{a} \in \mathcal{A}$ the following inequality holds:

$$\hat{\boldsymbol{M}} Q_S(s, \boldsymbol{a}, g | \cdot) - \min_{\boldsymbol{a} \in \mathcal{A}} \left( -\mathcal{R}_S(s, \boldsymbol{a}, g) + \gamma \mathcal{P}^{\boldsymbol{a}}_{s' s} v'_S(s') \right) < 0. \tag{13}$$

We now observe the following:

$$\hat{M}Q_S(s,\boldsymbol{a},g|\cdot) - \min_{\boldsymbol{a}\in\mathcal{A}}\left(-\mathcal{R}_S(s,\boldsymbol{a},g) + \gamma\mathcal{P}^{\boldsymbol{a}}_{s's}v'_S(s')\right)$$

$$\leq \max\left\{\min_{\boldsymbol{a}\in\mathcal{A}}\left(-\mathcal{R}_S(s,\boldsymbol{a},g) + \gamma\mathcal{P}^{\boldsymbol{a}}_{s's}v_S(s')\right), \hat{M}Q_S(s,\boldsymbol{a},g|\cdot)\right\} - \min_{\boldsymbol{a}\in\mathcal{A}}\left(-\mathcal{R}_S(s,\boldsymbol{a},g) + \gamma\mathcal{P}^{\boldsymbol{a}}_{s's}v'_S(s')\right)$$

$$\leq \left| \max\left\{\min_{\boldsymbol{a}\in\mathcal{A}}\left(-\mathcal{R}_S(s,\boldsymbol{a},g) + \gamma\mathcal{P}^{\boldsymbol{a}}_{s's}v_S(s')\right), \hat{M}Q_S(s,\boldsymbol{a},g|\cdot)\right\} \right.$$

$$- \max\left\{\min_{\boldsymbol{a}\in\mathcal{A}}\left(-\mathcal{R}_S(s,\boldsymbol{a},g) + \gamma\mathcal{P}^{\boldsymbol{a}}_{s's}v'_S(s')\right), \hat{M}Q_S(s,\boldsymbol{a},g|\cdot)\right\}$$

$$\left. + \max\left\{\min_{\boldsymbol{a}\in\mathcal{A}}\left(-\mathcal{R}_S(s,\boldsymbol{a},g) + \gamma\mathcal{P}^{\boldsymbol{a}}_{s's}v'_S(s')\right), \hat{M}Q_S(s,\boldsymbol{a},g|\cdot)\right\} - \min_{\boldsymbol{a}\in\mathcal{A}}\left(-\mathcal{R}_S(s,\boldsymbol{a},g) + \gamma\mathcal{P}^{\boldsymbol{a}}_{s's}v'_S(s')\right)\right|$$

$$\leq \left| \max\left\{\min_{\boldsymbol{a}\in\mathcal{A}}\left(-\mathcal{R}_S(s,\boldsymbol{a},g) + \gamma\mathcal{P}^{\boldsymbol{a}}_{s's}v_S(s')\right), \hat{M}Q_S(s,\boldsymbol{a},g|\cdot)\right\} \right.$$

$$\left. - \max\left\{\min_{\boldsymbol{a}\in\mathcal{A}}\left(-\mathcal{R}_S(s,\boldsymbol{a},g) + \gamma\mathcal{P}^{\boldsymbol{a}}_{s's}v'_S(s')\right), \hat{M}Q_S(s,\boldsymbol{a},g|\cdot)\right\}\right|$$

$$+ \left| \max\left\{\min_{\boldsymbol{a}\in\mathcal{A}}\left(-\mathcal{R}_S(s,\boldsymbol{a},g) + \gamma\mathcal{P}^{\boldsymbol{a}}_{s's}v'_S(s')\right), \hat{M}Q_S(s,\boldsymbol{a},g|\cdot)\right\} - \min_{\boldsymbol{a}\in\mathcal{A}}\left(-\mathcal{R}_S(s,\boldsymbol{a},g) + \gamma\mathcal{P}^{\boldsymbol{a}}_{s's}v'_S(s')\right)\right|$$

$$\leq \gamma\max_{\boldsymbol{a}\in\mathcal{A}}|\mathcal{P}^{\boldsymbol{a}}_{s's}v_S(s') - \mathcal{P}^{\boldsymbol{a}}_{s's}v'_S(s')| + \left|\max\left\{0, \hat{M}Q_S(s,\boldsymbol{a},g|\cdot) - \min_{\boldsymbol{a}\in\mathcal{A}}\left(-\mathcal{R}_S(s,\boldsymbol{a},g) + \gamma\mathcal{P}^{\boldsymbol{a}}_{s's}v'_S(s')\right)\right\}\right|$$

$$\leq \gamma\|Pv_S - Pv'_S\|$$

$$\leq \gamma\|v_S - v'_S\|,$$

where we have used the fact that for any scalars $a, b, c$ we have that $|\max\{a,b\} - \max\{b,c\}| \leq |a - c|$, the non-expansiveness of $P$ and Lemma 4.

**Case 2:** Let us now consider the case:

$$\hat{M}Q_S(s,\boldsymbol{a},g|\cdot) - \min_{\boldsymbol{a}\in\mathcal{A}}\left(-\mathcal{R}_S(s,\boldsymbol{a},g) + \gamma\mathcal{P}^{\boldsymbol{a}}_{s's}v'_S(s')\right) \geq 0.$$

For this case, first recall that $c > 0$, hence

$$\hat{M}Q_S(s,\boldsymbol{a},g|\cdot) - \min_{\boldsymbol{a}\in\mathcal{A}}\left(-\mathcal{R}_S(s,\boldsymbol{a},g) + \gamma\mathcal{P}^{\boldsymbol{a}}_{s's}v'_S(s')\right)$$

$$\leq \hat{M}Q_S(s,\boldsymbol{a},g|\cdot) - \min_{\boldsymbol{a}\in\mathcal{A}}\left(-\mathcal{R}_S(s,\boldsymbol{a},g) + \gamma\mathcal{P}^{\boldsymbol{a}}_{s's}v'_S(s') - c\right)$$

$$= \max_{i\in\mathcal{N},a^i\in\mathcal{A}^i}\min_{a^{-i}\in\mathcal{A}^{-i}}\left(-\mathcal{R}_S(s,\boldsymbol{a},g) - c + \gamma\mathcal{P}^{a^i}_{s's}\mathcal{P}^{a^{-i}}_{s's}v_S(s')\right) - \min_{\boldsymbol{a}\in\mathcal{A}}\left(-\mathcal{R}_S(s,\boldsymbol{a},g) - c + \gamma\mathcal{P}^{\boldsymbol{a}}_{s's}v'_S(s')\right)$$

$$\leq \max_{i\in\mathcal{N},a^i\in\mathcal{A}^i}\max_{a^{-i}\in\mathcal{A}^{-i}}\left(-\mathcal{R}_S(s,\boldsymbol{a},g) + \gamma\mathcal{P}^{a^i}_{s's}\mathcal{P}^{a^{-i}}_{s's}v_S(s')\right) - \min_{\boldsymbol{a}\in\mathcal{A}}\left(-\mathcal{R}_S(s,\boldsymbol{a},g) + \gamma\mathcal{P}^{\boldsymbol{a}}_{s's}v'_S(s')\right)$$

$$= \max_{i\in\mathcal{N},a^i\in\mathcal{A}^i}\max_{a^{-i}\in\mathcal{A}^{-i}}\left(-\mathcal{R}_S(s,\boldsymbol{a},g) + \gamma\mathcal{P}^{a^i}_{s's}\mathcal{P}^{a^{-i}}_{s's}v_S(s')\right) + \max_{\boldsymbol{a}\in\mathcal{A}}\left(\mathcal{R}_S(s,\boldsymbol{a},g) - \gamma\mathcal{P}^{\boldsymbol{a}}_{s's}v'_S(s')\right)$$

$$\leq \max_{\boldsymbol{a}\in\mathcal{A}}\left(-\mathcal{R}_S(s,\boldsymbol{a},g) + \gamma\mathcal{P}^{\boldsymbol{a}}_{s's}v_S(s')\right) + \max_{\boldsymbol{a}\in\mathcal{A}}\left(\mathcal{R}_S(s,\boldsymbol{a},g) - \gamma\mathcal{P}^{\boldsymbol{a}}_{s's}v'_S(s')\right)$$

$$\leq \gamma\max_{\boldsymbol{a}\in\mathcal{A}}|\mathcal{P}^{\boldsymbol{a}}_{s's}\left(v_S(s') - v'_S(s')\right)|$$

$$\leq \gamma|v_S(s') - v'_S(s')|$$

$$\leq \gamma\|v_S - v'_S\|,$$

using the non-expansiveness of the operator $P$ and Lemma 4. Hence we have that

$$\left\|\hat{M}Q_S - \min_{\boldsymbol{a}\in\mathcal{A}}\left[-\mathcal{R}_S(\cdot,\boldsymbol{a}) + \gamma\mathcal{P}^{\boldsymbol{a}}_{s's}v'_S\right]\right\| \leq \gamma\|v_S - v'_S\|. \tag{14}$$

Gathering the results of the three cases completes the proof of Theorem 2.

PROOF OF THEOREM 1

The proposition is proven by proving three auxiliary results:

**Lemma 7.** *For any $v_S : \mathcal{S} \to \mathbb{R}$, the solution of the game $\mathcal{G}$ is a limit point of the convergent sequence $T^1 v_S, T^2 v_S, \ldots$.*

*Proof of Lemma 7.* By Lemma 6, we have that

$$\|T^{k+1} v_S - T^k v_S\| \le \gamma \|T^k v_S - T^{k-1} v_S\| \le \cdots \le \gamma^k \|T v_S - v_S\|. \tag{15}$$

The result follows after considering the limit as $k \to \infty$ and using the boundedness of $\|T v_S - v_S\|$, we deduce that $T^k v_S, T^{k+1} v_S, \ldots$, is a Cauchy sequence which concludes the proof. $\square$

**Proposition 3.** *Denote by the finite game $\mathcal{G}^k$ in which Switcher, Adversary and $N$ MARL agents seek to maximise the following finite-horizon objectives:*

$$v_S^k(s|\boldsymbol{\pi}, \mathfrak{g}) = -\mathbb{E}_{\mathfrak{g}, \boldsymbol{\sigma}}\left[ \sum_{0 \le t \le k < \infty} \gamma^t \left( \mathcal{R}(s_t, \boldsymbol{a}_t) + c \cdot \mathbf{1}_{\mathcal{N}}(g(s_t)) \right) \Big| s_0 = s; \boldsymbol{a}_t \sim (\boldsymbol{\pi}, \boldsymbol{\sigma}) \right], \ \forall s \in \mathcal{S},$$

$$v_A^k(s|\boldsymbol{\pi}, \mathfrak{g}) = -\mathbb{E}_{\mathfrak{g}, \boldsymbol{\sigma}}\left[ \sum_{0 \le t \le k < \infty} \gamma^t \mathcal{R}(s_t, \boldsymbol{a}_t) \Big| s_0 = s; \boldsymbol{a}_t \sim (\boldsymbol{\pi}, \boldsymbol{\sigma}) \right], \ \forall s \in \mathcal{S},$$

$$v^k(s|\boldsymbol{\pi}, \mathfrak{g}) = \mathbb{E}_{\mathfrak{g}, \boldsymbol{\sigma}}\left[ \sum_{0 \le t \le k < \infty} \gamma^t \mathcal{R}(s_t, \boldsymbol{a}_t) \Big| s_0 = s; \boldsymbol{a}_t \sim (\boldsymbol{\pi}, \boldsymbol{\sigma}) \right], \ \forall s \in \mathcal{S},$$

*Let $\hat{\mathfrak{g}}$ and $\hat{\boldsymbol{\sigma}}, \hat{\boldsymbol{\pi}} \in \boldsymbol{\Pi}$ be the Markov policies generated by the process $T^1 v_S, T^2 v_S, \ldots$, then $\hat{\mathfrak{g}}$ and $\hat{\boldsymbol{\sigma}}, \hat{\boldsymbol{\pi}} \in \boldsymbol{\Pi}$ are optimal policies for $\mathcal{G}^k$.*

*Proof.* The proof is achieved using similar arguments as those presented in Theorem 2 of (Shapley, 1953) with some modifications. Denote by the *finite* game $\mathcal{G}^k$ of $k < \infty$ steps in which adversary (the $N$ agents) maximises (minimise) the following objective $v^k(s|\boldsymbol{\pi}, \mathfrak{g}) = -\mathbb{E}\left[ \sum_{t=0}^{k} \gamma^t \{ \mathcal{R}(s_t, a_t, b_t) \} \Big| s_0 = s \right]$ and Switcher maximises $v_C^k(s|\boldsymbol{\pi}, \mathfrak{g}) = -\mathbb{E}\left[ \sum_{t=0}^{k} \gamma^t \left( \mathcal{R}(s_t, \boldsymbol{a}_t) + c \cdot \mathbf{1}_{\mathcal{N}}(g(s_t)) \right) \Big| s_0 = s \right]$. Suppose in the game $\mathcal{G}^k$, Switcher is given a payoff of $-\mathcal{R}(s, \boldsymbol{a}) + \mathcal{P}_{s's}^{\boldsymbol{a}} v_S(s'|\boldsymbol{\pi}, \mathfrak{g})$ given Switcher policy $\mathfrak{g}$ and for any given $s \in \mathcal{S}$ and $\boldsymbol{a} \sim \boldsymbol{\pi}, \sigma$. Now the Markov policy $\mathfrak{g}$ guarantees Switcher a payoff of $v^k(s|\boldsymbol{\pi}, \mathfrak{g})$. Now in the game $\mathcal{G}^k$, after $n < \infty$ steps and using the policy $\mathfrak{g}$ gives Switcher an expected payoff of at least $v^k(s|\boldsymbol{\pi}, \mathfrak{g}) - \gamma^{n-1} \max_{\boldsymbol{a} \in \mathcal{A}} \mathcal{P}_{s's}^{\boldsymbol{a}} v^k(s'|\boldsymbol{\pi}, \mathfrak{g}) \le v^k(s|\boldsymbol{\pi}, \mathfrak{g}) - \gamma^{n-1} \max_{s' \in \mathcal{S}} v^k(s'|\boldsymbol{\pi}, \mathfrak{g})$. Therefore, accounting for the $n$ steps, the total payoff for Switcher is at least $v^k(s|\boldsymbol{\pi}, \mathfrak{g}) - \gamma^{n-1} \max_{s' \in \mathcal{S}} v(s'|\boldsymbol{\pi}, \mathfrak{g}) - \sum_{t=0}^{n-1} \gamma^t v^{n-t}(s'|\boldsymbol{\pi}, \mathfrak{g}) = v^k(s|\boldsymbol{\pi}, \mathfrak{g}) - \gamma^{n-1} \max_{s' \in \mathcal{S}} v(s'|\boldsymbol{\pi}, \mathfrak{g}) - \gamma^n \frac{1-\gamma^n}{1-\gamma} \|v\| := \tilde{\boldsymbol{v}}^{k,n}(s|\boldsymbol{\pi}, \mathfrak{g})$. This expression holds for arbitrarily large values of $n$ in particular $\lim_{n \to \infty} \tilde{\boldsymbol{v}}^{k,n}(s|\boldsymbol{\pi}, \mathfrak{g}) = v^k(s|\boldsymbol{\pi}, \mathfrak{g})$ from which it follows that the policy $\mathfrak{g}$ is optimal for Switcher. Now by Assumption 3 it is easy to see for any fixed $K < \infty$, any policy which is optimal for Switcher in the game $\mathcal{G}$ is optimal for Switcher in a game $\tilde{\mathcal{G}}$ in which Switcher's objective is $v_S^k(s|\boldsymbol{\pi}, \mathfrak{g}) = -\mathbb{E}_{\mathfrak{g}, \boldsymbol{\sigma}}\left[ \sum_{0 \le t \le k < \infty} \gamma^t \left( \mathcal{R}(s_t, \boldsymbol{a}_t) + c \cdot \mathbf{1}_{\mathcal{N}}(g(s_t)) \right) \Big| s_0 = s; \boldsymbol{a}_t \sim (\boldsymbol{\pi}, \boldsymbol{\sigma}) \right]$ with the game being otherwise identical. After using analogous arguments for Switcher and adversary, we deduce the result. $\square$

**Lemma 8.** *The value of the game $\mathcal{G}$ is unique.*

*Proof.* Suppose there exists two values the game $\mathcal{G}$, $v_S'$ and $v_S$. Then, since each is a solution, we have by Lemma 6 that each is a fixed point of the Bellman operator and hence $T v_S = v_S$ and $T v_S' = v_S'$. Hence, we have the following:

$$\|v_S - v_S'\| = \|T v_S - T v_S'\| \le \gamma \|v_S - v_S'\|, \tag{16}$$

whereafter, we immediately deduce that $v_S = v_S'$. $\square$

Summing up the above results we have succeeded in proving Theorem 1, moreover Proposition 1 follows as an immediate consequence. □

To prove the Theorem 3, we make use of the following result:

**Theorem 6** (Theorem 1, pg 4 in (Jaakkola et al., 1994)). *Let $\Xi_t(s)$ be a random process that takes values in $\mathbb{R}^n$ and given by the following:*

$$\Xi_{t+1}(s) = (1 - \alpha_t(s))\,\Xi_t(s)\alpha_t(s)L_t(s), \tag{17}$$

*then $\Xi_t(s)$ converges to $0$ with probability 1 under the following conditions:*

  *i)* $0 \leq \alpha_t \leq 1, \sum_t \alpha_t = \infty$ *and* $\sum_t \alpha_t < \infty$

  *ii)* $\|\mathbb{E}[L_t|\mathcal{F}_t]\| \leq \gamma\|\Xi_t\|$, *with* $\gamma < 1$;

  *iii)* $\mathrm{Var}\,[L_t|\mathcal{F}_t] \leq c(1 + \|\Xi_t\|^2)$ *for some* $c > 0$.

*Proof.* To prove the result, we show (i) - (iii) hold. Condition (i) holds by choice of learning rate. It therefore remains to prove (ii) - (iii). We first prove (ii). For this, we consider our variant of the Q-learning update rule:

$$\boldsymbol{Q}_{S,t+1}(s_t, \boldsymbol{a}_t, g|\cdot) = \boldsymbol{Q}_{S,t}(s_t, \boldsymbol{a}_t, g|\cdot)$$
$$+ \alpha_t(s_t, \boldsymbol{a}_t)\Big[\max\Big\{\hat{\boldsymbol{\mathcal{M}}}Q_S(s_{\tau_k}, \boldsymbol{a}, g|\cdot), \mathcal{R}(s_{\tau_k}, \boldsymbol{a}, g) + \gamma\max_{\boldsymbol{a}'\in\mathcal{A}}Q_S(s_{t+1}, \boldsymbol{a}', g|\cdot)\Big\}$$
$$- \boldsymbol{Q}_t(s_t, \boldsymbol{a}_t, g|\cdot)\Big].$$

After subtracting $\boldsymbol{Q}_S^*(s_t, \boldsymbol{a}_t, g|\cdot)$ from both sides and some manipulation we obtain that:

$$\Xi_{t+1}(s_t, \boldsymbol{a}_t)$$
$$= (1 - \alpha_t(s_t, \boldsymbol{a}_t))\Xi_t(s_t, \boldsymbol{a}_t)$$
$$+ \alpha_t(s_t, \boldsymbol{a}_t))\Big[\max\Big\{\hat{\boldsymbol{\mathcal{M}}}Q_S(s_{\tau_k}, \boldsymbol{a}, g|\cdot), \mathcal{R}_S(s_{\tau_k}, \boldsymbol{a}, g) + \gamma\max_{\boldsymbol{a}'\in\mathcal{A}}\boldsymbol{Q}_S(s', \boldsymbol{a}', g|\cdot)\Big\} - \boldsymbol{Q}_S^*(s_t, \boldsymbol{a}_t, g|\cdot)\Big],$$

where $\Xi_t(s_t, \boldsymbol{a}_t, g) := \boldsymbol{Q}_{S,t}(s_t, \boldsymbol{a}_t, g|\cdot) - Q_S^*(s_t, \boldsymbol{a}_t, g|\cdot)$.

Let us now define by

$$L_t(s_{\tau_k}, \boldsymbol{a}, g) := \max\Big\{\hat{\boldsymbol{\mathcal{M}}}Q_S(s_{\tau_k}, \boldsymbol{a}, g|\cdot), \mathcal{R}_S(s_{\tau_k}, \boldsymbol{a}, g) + \gamma\max_{\boldsymbol{a}'\in\mathcal{A}}\boldsymbol{Q}_S(s', \boldsymbol{a}', g|\cdot)\Big\} - \boldsymbol{Q}_S^*(s_t, \boldsymbol{a}, g|\cdot).$$

Then

$$\Xi_{t+1}(s_t, \boldsymbol{a}_t, g) = (1 - \alpha_t(s_t, \boldsymbol{a}_t))\Xi_t(s_t, \boldsymbol{a}_t, g) + \alpha_t(s_t, \boldsymbol{a}_t))\,[L_t(s_{\tau_k}, \boldsymbol{a}, g)]. \tag{18}$$

We now observe that

$$\mathbb{E}\,[L_t(s_{\tau_k}, \boldsymbol{a}, g)|\mathcal{F}_t]$$
$$= \sum_{s'\in\mathcal{S}} P(s'; a, s_{\tau_k})\max\Big\{\hat{\boldsymbol{\mathcal{M}}}Q_S(s_{\tau_k}, \boldsymbol{a}, g|\cdot), \mathcal{R}_S(s_{\tau_k}, \boldsymbol{a}, g) + \gamma\max_{\boldsymbol{a}'\in\mathcal{A}}\boldsymbol{Q}_S(s', \boldsymbol{a}', g|\cdot)\Big\}$$
$$- \boldsymbol{Q}_S^*(s_{\tau_k}, a, g|\cdot)$$
$$= T\boldsymbol{Q}_{S,t}(s, \boldsymbol{a}, g|\cdot) - \boldsymbol{Q}_S^*(s, \boldsymbol{a}, g). \tag{19}$$

Now, using the fixed point property that implies $\boldsymbol{Q}^* = T\boldsymbol{Q}^*$, we find that

$$\mathbb{E}\,[L_t(s_{\tau_k}, \boldsymbol{a}, g)|\mathcal{F}_t] = T\boldsymbol{Q}_{S,t}(s, \boldsymbol{a}, g|\cdot) - T\boldsymbol{Q}_S^*(s, \boldsymbol{a}, g|\cdot)$$
$$\leq \|T\boldsymbol{Q}_{S,t} - T\boldsymbol{Q}^*\|$$
$$\leq \gamma\,\|\boldsymbol{Q}_{S,t} - \boldsymbol{Q}^*\|_\infty = \gamma\,\|\Xi_t\|_\infty. \tag{20}$$

using the contraction property of $T$ established in Lemma 6. This proves (ii).

We now prove iii), that is

$$\mathrm{Var}\left[L_t | \mathcal{F}_t\right] \le c(1 + \|\Xi_t\|^2). \tag{21}$$

Now by equation 19 we have that

$$\mathrm{Var}\left[L_t | \mathcal{F}_t\right] = \mathrm{Var}\left[\max\left\{\hat{\mathcal{M}}Q_S(s_{\tau_k}, \boldsymbol{a}, g|\cdot), \mathcal{R}_S(s_{\tau_k}, \boldsymbol{a}, g) + \gamma \max_{\boldsymbol{a}' \in \mathcal{A}} \boldsymbol{Q}_S(s', \boldsymbol{a}', g|\cdot)\right\} - \boldsymbol{Q}_S^*(s_t, \boldsymbol{a}, g|\cdot)\right]$$

$$= \mathbb{E}\left[\left(\max\left\{\hat{\mathcal{M}}Q_S(s_{\tau_k}, \boldsymbol{a}, g|\cdot), \mathcal{R}_S(s_{\tau_k}, \boldsymbol{a}, g) + \gamma \max_{\boldsymbol{a}' \in \mathcal{A}} \boldsymbol{Q}_S(s', \boldsymbol{a}', g|\cdot)\right\}\right.\right.$$

$$\left.\left. - \boldsymbol{Q}_S^*(s_t, \boldsymbol{a}, g|\cdot) - (T\boldsymbol{Q}_{S,t}(s, \boldsymbol{a}, g|\cdot) - \boldsymbol{Q}_S^*(s, \boldsymbol{a}, g|\cdot))\right)^2\right]$$

$$= \mathbb{E}\left[\left(\max\left\{\hat{\mathcal{M}}Q_S(s_{\tau_k}, \boldsymbol{a}, g|\cdot), \mathcal{R}_S(s_{\tau_k}, \boldsymbol{a}, g) + \gamma \max_{\boldsymbol{a}' \in \mathcal{A}} Q_S(s', \boldsymbol{a}', g|\cdot)\right\} - T\boldsymbol{Q}_{S,t}(s, \boldsymbol{a}, g|\cdot)\right)^2\right]$$

$$= \mathrm{Var}\left[\max\left\{\hat{\mathcal{M}}Q_S(s_{\tau_k}, \boldsymbol{a}, g|\cdot), \mathcal{R}_S(s_{\tau_k}, \boldsymbol{a}, g) + \gamma \max_{\boldsymbol{a}' \in \mathcal{A}} Q_S(s', \boldsymbol{a}', g|\cdot)\right\} - T\boldsymbol{Q}_{S,t}(s, \boldsymbol{a}, g|\cdot))\right]$$

$$\le c(1 + \|\Xi_t\|^2),$$

for some $c > 0$ where the last line follows due to the boundedness of $Q$ (which follows from Assumptions 2 and 4). This concludes the proof of the Theorem. $\qquad\square$

Theorem 3 proves the convergence of MARTA to the solution to the game $\mathcal{G}$ (and that such a solution exists). In particular, the following result follows a consequence of Theorem 3 and the uniqueness of the value function solution of $\mathcal{G}$:

# E  CONVERGENCE OF MARTA WITH LINEAR FUNCTION APPROXIMATORS

In reinforcement learning, an important consideration is the use of function approximators for the functions being learned during the learning process. We now extend the convergence results established in the previous section to linear function approximators. Linear function approximators are an important class due to their simplicity and convexity properties that do not suffer from issues such as convergence to suboptimal stationary points. Moreover, the following analysis is an important stepping stone for proving analogous results with other function approximator classes.

In addition to Assumptions 1 - 3, the results of this section are built under the following assumptions:

Assumption 4. For any positive scalar $c$, there exists a scalar $\kappa_c$ such that for all $s \in \mathcal{S}$ and for any $t \in \mathbb{N}$ we have: $\mathbb{E}\left[1 + \|s_t\|^c | s_0 = s\right] \le \kappa_c(1 + \|s\|^c)$.

Assumption 5. There exists scalars $C_1$ and $c_1$ such that for any function $v$ satisfying $|v(s)| \le C_2(1 + \|s\|^{c_2})$ for some scalars $c_2$ and $C_2$ we have that: $\sum_{t=0}^{\infty} |\mathbb{E}\left[v(s_t) | s_0 = s\right] - \mathbb{E}[v(s_0)]| \le C_1 C_2(1 + \|s_0\|^{c_1 c_2})$.

Assumption 6. There exists scalars $c$ and $C$ such that for any $s \in \mathcal{S}$ we have that $|R(s, \cdot)| \le C(1 + \|s\|^c)$.

Theorem 4 is proven using a set of results that we now establish. First we prove the following bound holds:

**Lemma 9.** *For any $\boldsymbol{Q} \in L_2$ we have that*

$$\|\mathfrak{F}\boldsymbol{Q} - \boldsymbol{Q}'\| \le \gamma \|\boldsymbol{Q} - \boldsymbol{Q}'\|, \tag{22}$$

*so that the operator $\mathfrak{F}$ is a contraction.*

*Proof.* Now, we first note that by result iv) in the proof of Lemma 6, we deduced that for any $\boldsymbol{Q}, v \in L_2$ we have that

$$\|\mathcal{M}\boldsymbol{Q} - [\mathcal{R}(\cdot, \boldsymbol{a}) + \gamma \mathcal{P}^{\boldsymbol{a}} v']\| \le \gamma \|v - v'\|.$$

where $\mathcal{P}^{\boldsymbol{a}}_{ss'} =: \sum_{s' \in \mathcal{S}} P(s'; \boldsymbol{a}, s)$.

Hence, using the contraction property of $\mathcal{M}$ and results i)-iv) of Lemma 6, we readily deduce the following bound:

$$\max \left\{ \left\| \mathcal{M}\boldsymbol{Q} - \hat{\boldsymbol{Q}} \right\|, \left\| \mathcal{M}\boldsymbol{Q} - \mathcal{M}\hat{\boldsymbol{Q}} \right\| \right\} \le \gamma \left\| \boldsymbol{Q} - \hat{\boldsymbol{Q}} \right\|. \tag{23}$$

We now observe that $\mathfrak{F}$ is a contraction. Indeed, since for any $\boldsymbol{Q}, \boldsymbol{Q}' \in L_2$ we have that:

$$\begin{aligned}
&\| \mathfrak{F}\boldsymbol{Q} - \mathfrak{F}\boldsymbol{Q}' \| \\
&= \left\| \mathcal{R} + \gamma P \min\{\hat{\mathcal{M}}Q, Q\} - \left( \mathcal{R} + \gamma P \min\{\hat{\mathcal{M}}Q', Q'\} \right) \right\| \\
&= \gamma \left\| P \min\{\hat{\mathcal{M}}Q, Q\} - P \min\{\hat{\mathcal{M}}Q', Q'\} \right\| \\
&\le \gamma \max \left\{ \| \boldsymbol{Q} - \boldsymbol{Q}' \|, \gamma \| \boldsymbol{Q} - \boldsymbol{Q}' \| \right\} \\
&= \gamma \| \boldsymbol{Q} - \boldsymbol{Q}' \|
\end{aligned}$$

using the Cauchy-Schwarz inequality, equation 23 and again using the non-expansiveness of $P$. $\quad\square$

We next show that the following two bounds hold:

**Lemma 10.** *For any $\boldsymbol{Q} \in \mathcal{V}$ we have that*

$$i) \qquad \left\| \Pi\mathfrak{F}\boldsymbol{Q} - \Pi\mathfrak{F}\bar{\boldsymbol{Q}} \right\| \le \gamma \left\| \boldsymbol{Q} - \bar{\boldsymbol{Q}} \right\|,$$

$$ii) \qquad \| \Phi r^\star - \boldsymbol{Q}^\star \| \le \frac{1}{\sqrt{1-\gamma^2}} \| \Pi\boldsymbol{Q}^\star - \boldsymbol{Q}^\star \|.$$

*Proof.* The first result is straightforward since as $\Pi$ is a projection it is non-expansive and hence:

$$\left\| \Pi\mathfrak{F}\boldsymbol{Q} - \Pi\mathfrak{F}\bar{\boldsymbol{Q}} \right\| \le \left\| \mathfrak{F}\boldsymbol{Q} - \mathfrak{F}\bar{\boldsymbol{Q}} \right\| \le \gamma \left\| \boldsymbol{Q} - \bar{\boldsymbol{Q}} \right\|,$$

using the contraction property of $\mathfrak{F}$. This proves i). For ii), we note that by the orthogonality property of projections we have that $\langle \Phi r^\star - \Pi\boldsymbol{Q}^\star, \Phi r^\star - \Pi\boldsymbol{Q}^\star \rangle = 0$, hence we observe that:

$$\begin{aligned}
\| \Phi r^\star - \boldsymbol{Q}^\star \|^2 &= \| \Phi r^\star - \Pi\boldsymbol{Q}^\star \|^2 + \| \boldsymbol{Q}^* - \Pi\boldsymbol{Q}^\star \|^2 \\
&= \| \Pi\mathfrak{F}\Phi r^\star - \Pi\boldsymbol{Q}^\star \|^2 + \| \boldsymbol{Q}^* - \Pi\boldsymbol{Q}^\star \|^2 \\
&\le \| \mathfrak{F}\Phi r^\star - \boldsymbol{Q}^\star \|^2 + \| \boldsymbol{Q}^* - \Pi\boldsymbol{Q}^\star \|^2 \\
&= \| \mathfrak{F}\Phi r^\star - \mathfrak{F}\boldsymbol{Q}^\star \|^2 + \| \boldsymbol{Q}^* - \Pi\boldsymbol{Q}^\star \|^2 \\
&\le \gamma^2 \| \Phi r^\star - \boldsymbol{Q}^\star \|^2 + \| \boldsymbol{Q}^* - \Pi\boldsymbol{Q}^\star \|^2,
\end{aligned}$$

after which we readily deduce the desired result. $\quad\square$

**Lemma 11.** *Define the operator $H$ by the following:*

$$H\boldsymbol{Q}(s, a, b) = \begin{cases} \hat{\mathcal{M}}\boldsymbol{Q}(s, \boldsymbol{a}), & \text{if } \hat{\mathcal{M}}\boldsymbol{Q}(s, \boldsymbol{a}) < \Phi r^\star \\ \boldsymbol{Q}(s, \boldsymbol{a}), & \text{otherwise}, \end{cases}$$

*where we define $\tilde{\mathfrak{F}}$ by: $\tilde{\mathfrak{F}}\boldsymbol{Q} := \mathcal{R} + \gamma PH\boldsymbol{Q}$.*

*For any $\boldsymbol{Q}, \bar{\boldsymbol{Q}} \in L_2$ we have that*

$$\left\| \tilde{\mathfrak{F}}\boldsymbol{Q} - \tilde{\mathfrak{F}}\bar{\boldsymbol{Q}} \right\| \le \gamma \left\| \boldsymbol{Q} - \bar{\boldsymbol{Q}} \right\| \tag{24}$$

*and hence $\tilde{\mathfrak{F}}$ is a contraction mapping.*

*Proof.* Using equation 23, we now observe that

$$\left\| \tilde{\mathfrak{F}} \boldsymbol{Q} - \tilde{\mathfrak{F}} \bar{\boldsymbol{Q}} \right\| = \left\| \mathcal{R} + \gamma P H \boldsymbol{Q} - \left( \mathcal{R} + \gamma P H \bar{\boldsymbol{Q}} \right) \right\|$$

$$\leq \gamma \left\| H \boldsymbol{Q} - H \bar{\boldsymbol{Q}} \right\|$$

$$\leq \gamma \left\| \max \left\{ \hat{\mathcal{M}} \boldsymbol{Q} - \hat{\mathcal{M}} \bar{\boldsymbol{Q}}, \hat{\mathcal{M}} \boldsymbol{Q} - \bar{\boldsymbol{Q}}, \hat{\mathcal{M}} \bar{\boldsymbol{Q}} - \boldsymbol{Q} \right\} \right\|$$

$$\leq \gamma \max \left\{ \left\| \hat{\mathcal{M}} \boldsymbol{Q} - \hat{\mathcal{M}} \bar{\boldsymbol{Q}} \right\|, \left\| \hat{\mathcal{M}} \boldsymbol{Q} - \bar{\boldsymbol{Q}} \right\|, \left\| \hat{\mathcal{M}} \bar{\boldsymbol{Q}} - \boldsymbol{Q} \right\| \right\}$$

$$\leq \gamma \max \left\{ \gamma \left\| \boldsymbol{Q} - \bar{\boldsymbol{Q}} \right\|, \left\| \boldsymbol{Q} - \bar{\boldsymbol{Q}} \right\| \right\}$$

$$= \gamma \left\| \boldsymbol{Q} - \bar{\boldsymbol{Q}} \right\|,$$

again using equation 23 and the non-expansive property of $P$. $\qquad \square$

**Lemma 12.** *Define by $\tilde{\boldsymbol{Q}} := \mathcal{R} + \gamma P v^{\tilde{\sigma}}$ where*

$$v^{\tilde{\sigma}}(s) := \min \left\{ \hat{\mathcal{M}} \boldsymbol{Q}^{\tilde{\sigma}}(s, \boldsymbol{a}), \mathcal{R}(s, \boldsymbol{a}) + \gamma \mathbb{E}_{s' \sim P} \left[ v^{\tilde{\sigma}}(s') \right] \right\}, \tag{25}$$

*then $\tilde{\boldsymbol{Q}}$ is a fixed point of $\tilde{\mathfrak{F}} \tilde{\boldsymbol{Q}}$, that is $\tilde{\mathfrak{F}} \tilde{\boldsymbol{Q}} = \tilde{\boldsymbol{Q}}$.*

*Proof.* We begin by observing that

$$H \tilde{\boldsymbol{Q}}(s, \boldsymbol{a}) = H \left( \mathcal{R}(s, \boldsymbol{a}) + \gamma \mathcal{P}_{ss'}^{\boldsymbol{a}} v^{\tilde{\sigma}}(s') \right)$$

$$= \begin{cases} \hat{\mathcal{M}} \boldsymbol{Q}(s, \boldsymbol{a}), & \text{if } \hat{\mathcal{M}} \boldsymbol{Q}(s, \boldsymbol{a}) > \Phi r^{\star} \\ \boldsymbol{Q}(s, \boldsymbol{a}), & \text{otherwise,} \end{cases}$$

$$= \begin{cases} \hat{\mathcal{M}} \boldsymbol{Q}(s, \boldsymbol{a}), & \text{if } \hat{\mathcal{M}} \boldsymbol{Q}(s, \boldsymbol{a}) > \Phi r^{\star}, \\ \mathcal{R}(s, \boldsymbol{a}) + \gamma P v^{\tilde{\sigma}}, & \text{otherwise,} \end{cases}$$

$$= v^{\tilde{\sigma}}(s).$$

Hence,

$$\tilde{\mathfrak{F}} \tilde{\boldsymbol{Q}} = \mathcal{R} + \gamma P H \tilde{\boldsymbol{Q}} = \mathcal{R} + \gamma P v^{\tilde{\sigma}} = \tilde{\boldsymbol{Q}}. \tag{26}$$

which proves the result. $\qquad \square$

**Lemma 13.** *The following bound holds:*

$$\mathbb{E} \left[ v^{\hat{\sigma}}(s_0) \right] - \mathbb{E} \left[ v^{\tilde{\sigma}}(s_0) \right] \leq 2 \left[ (1 - \gamma) \sqrt{(1 - \gamma^2)} \right]^{-1} \left\| \Pi \boldsymbol{Q}^{\star} - \boldsymbol{Q}^{\star} \right\|. \tag{27}$$

*Proof.* By definitions of $v^{\hat{\sigma}}$ and $v^{\tilde{\sigma}}$ (c.f equation 25) and using Jensen's inequality and the stationarity property we have that,

$$\mathbb{E} \left[ v^{\hat{\sigma}}(s_0) \right] - \mathbb{E} \left[ v^{\tilde{\sigma}}(s_0) \right] = \mathbb{E} \left[ P v^{\hat{\sigma}}(s_0) \right] - \mathbb{E} \left[ P v^{\tilde{\sigma}}(s_0) \right]$$

$$\leq \left| \mathbb{E} \left[ P v^{\hat{\sigma}}(s_0) \right] - \mathbb{E} \left[ P v^{\tilde{\sigma}}(s_0) \right] \right|$$

$$\leq \left\| P v^{\hat{\sigma}} - P v^{\tilde{\sigma}} \right\|. \tag{28}$$

Now recall that $\tilde{\boldsymbol{Q}} := \mathcal{R} + \gamma P v^{\tilde{\sigma}}$ and $\boldsymbol{Q}^{\star} := \mathcal{R} + \gamma P v^{\sigma^{\star}}$, using these expressions in equation 28 we find that

$$\mathbb{E} \left[ v^{\hat{\sigma}}(s_0) \right] - \mathbb{E} \left[ v^{\tilde{\sigma}}(s_0) \right] \leq \frac{1}{\gamma} \left\| \tilde{\boldsymbol{Q}} - \boldsymbol{Q}^{\star} \right\|.$$

Moreover, by the triangle inequality and using the fact that $\mathfrak{F}(\Phi r^{\star}) = \tilde{\mathfrak{F}}(\Phi r^{\star})$ and that $\mathfrak{F} \boldsymbol{Q}^{\star} = \boldsymbol{Q}^{\star}$ and $\tilde{\mathfrak{F}} \tilde{\boldsymbol{Q}} = \tilde{\boldsymbol{Q}}$ (c.f. equation 27) we have that

$$\left\| \tilde{\boldsymbol{Q}} - \boldsymbol{Q}^{\star} \right\| \leq \left\| \tilde{\boldsymbol{Q}} - \mathfrak{F}(\Phi r^{\star}) \right\| + \left\| \boldsymbol{Q}^{\star} - \tilde{\mathfrak{F}}(\Phi r^{\tilde{\sigma}}) \right\|$$

$$\leq \gamma \left\| \tilde{\boldsymbol{Q}} - \Phi r^{\star} \right\| + \gamma \left\| \boldsymbol{Q}^{\star} - \Phi r^{\star} \right\|$$

$$\leq 2\gamma \left\| \tilde{\boldsymbol{Q}} - \Phi r^{\star} \right\| + \gamma \left\| \boldsymbol{Q}^{\star} - \tilde{\boldsymbol{Q}} \right\|,$$

which gives the following bound:

$$\left\| \tilde{\boldsymbol{Q}} - \boldsymbol{Q}^\star \right\| \leq 2\gamma \left(1 - \gamma\right)^{-1} \left\| \tilde{\boldsymbol{Q}} - \Phi r^\star \right\|,$$

from which, using Lemma 10, we deduce that $\left\| \tilde{\boldsymbol{Q}} - \boldsymbol{Q}^\star \right\| \leq$ $2\gamma \left[(1-\gamma)\sqrt{(1-\gamma^2)}\right]^{-1} \|\Pi \boldsymbol{Q}^\star - \boldsymbol{Q}^\star\|$, after which by equation 29, we finally obtain

$$\mathbb{E}\left[v^{\hat{\boldsymbol{\sigma}}}(s_0)\right] - \mathbb{E}\left[v^{\tilde{\boldsymbol{\sigma}}}(s_0)\right] \leq 2 \left[(1-\gamma)\sqrt{(1-\gamma^2)}\right]^{-1} \|\Pi \boldsymbol{Q}^\star - \boldsymbol{Q}^\star\|,$$

as required. $\qquad\qquad\qquad\qquad\qquad\qquad\qquad\qquad\qquad\qquad\qquad\qquad\qquad\qquad\qquad\square$

Let us rewrite the update in the following way:

$$r_{t+1} = r_t + \gamma_t \Xi(w_t, r_t),$$

where the function $\Xi : \mathbb{R}^{2d} \times \mathbb{R}^p \to \mathbb{R}^p$ is given by:

$$\Xi(w, r) := \phi(s)\left(\mathcal{R}(s, \cdot) + \gamma \min\{(\Phi r)(s'), \hat{\boldsymbol{\mathcal{M}}}(\Phi r)(s')\} - (\Phi r)(s)\right),$$

for any $w \equiv (s, s') \in \mathcal{S}^2$ and for any $r \in \mathbb{R}^p$. Let us also define the function $\boldsymbol{\Xi} : \mathbb{R}^p \to \mathbb{R}^p$ by the following:

$$\boldsymbol{\Xi}(r) := \mathbb{E}_{w_0 \sim (\mathbb{P}, \mathbb{P})}\left[\Xi(w_0, r)\right]; w_0 := (s_0, z_1).$$

**Lemma 14.** *The following statements hold for all $z \in \{0, 1\} \times \mathcal{S}$:*

*i)* $(r - r^\star)\boldsymbol{\Xi}_k(r) < 0, \qquad \forall r \neq r^\star,$

*ii)* $\boldsymbol{\Xi}_k(r^\star) = 0.$

*Proof.* To prove the statement, we first note that each component of $\boldsymbol{\Xi}_k(r)$ admits a representation as an inner product, indeed:

$$\begin{aligned}
\boldsymbol{\Xi}_k(r) &= \mathbb{E}\left[\phi_k(s_0)(\mathcal{R}(s_0, \boldsymbol{a}_0) + \gamma \min\{(\Phi r)(s_1), \hat{\boldsymbol{\mathcal{M}}}(\Phi r)(s_1)\} - (\Phi r)(s_0)\right] \\
&= \mathbb{E}\left[\phi_k(s_0)(\mathcal{R}(s_0, \boldsymbol{a}_0) + \gamma \mathbb{E}\left[\min\{(\Phi r)(s_1), \hat{\boldsymbol{\mathcal{M}}}(\Phi r)(s_1)\}|x_0\right] - (\Phi r)(s_0)\right] \\
&= \mathbb{E}\left[\phi_k(s_0)(\mathcal{R}(s_0, \boldsymbol{a}_0) + \gamma P \min\{(\Phi r), \hat{\boldsymbol{\mathcal{M}}}(\Phi r)\}(s_0) - (\Phi r)(s_0)\right] \\
&= \langle \phi_k, \mathfrak{F}\Phi r - \Phi r \rangle,
\end{aligned}$$

using the iterated law of expectations and the definitions of $P$ and $\mathfrak{F}$.

We now are in a position to prove (i). Indeed, we now observe the following:

$$\begin{aligned}
(r - r^\star) \boldsymbol{\Xi}_k(r) &= \sum_{l=1} (r(l) - r^\star(l)) \langle \phi_l, \mathfrak{F}\Phi r - \Phi r \rangle \\
&= \langle \Phi r - \Phi r^\star, \mathfrak{F}\Phi r - \Phi r \rangle \\
&= \langle \Phi r - \Phi r^\star, (\boldsymbol{1} - \Pi)\mathfrak{F}\Phi r + \Pi\mathfrak{F}\Phi r - \Phi r \rangle \\
&= \langle \Phi r - \Phi r^\star, \Pi\mathfrak{F}\Phi r - \Phi r \rangle,
\end{aligned}$$

where in the last step we used the orthogonality of $(\boldsymbol{1} - \Pi)$. We now recall that $\Pi\mathfrak{F}\Phi r^\star = \Phi r^\star$ since $\Phi r^\star$ is a fixed point of $\Pi\mathfrak{F}$. Additionally, using Lemma 10 we observe that $\|\Pi\mathfrak{F}\Phi r - \Phi r^\star\| \leq \gamma \|\Phi r - \Phi r^\star\|$. With this we now find that

$$\begin{aligned}
&\langle \Phi r - \Phi r^\star, \Pi\mathfrak{F}\Phi r - \Phi r \rangle \\
&= \langle \Phi r - \Phi r^\star, (\Pi\mathfrak{F}\Phi r - \Phi r^\star) + \Phi r^\star - \Phi r \rangle \\
&\leq \|\Phi r - \Phi r^\star\| \|\Pi\mathfrak{F}\Phi r - \Phi r^\star\| - \|\Phi r^\star - \Phi r\|^2 \\
&\leq (\gamma - 1) \|\Phi r^\star - \Phi r\|^2,
\end{aligned}$$

which is negative since $\gamma < 1$ which completes the proof of part i).

The proof of part ii) is straightforward since we readily observe that

$$\Xi_k(r^\star) = \langle \phi_l, \mathfrak{F}\Phi r^\star - \Phi r \rangle = \langle \phi_l, \Pi \mathfrak{F}\Phi r^\star - \Phi r \rangle = 0,$$

as required and from which we deduce the result. $\qquad \square$

To prove the theorem, we make use of a special case of the following result:

**Theorem 7** (Th. 17, p. 239 in (Benveniste et al., 2012))**.** *Consider a stochastic process* $r_t :$ $\mathbb{R} \times \{\infty\} \times \Omega \to \mathbb{R}^k$ *which takes an initial value* $r_0$ *and evolves according to the following:*

$$r_{t+1} = r_t + \alpha \Xi(s_t, r_t), \tag{29}$$

*for some function* $s : \mathbb{R}^{2d} \times \mathbb{R}^k \to \mathbb{R}^k$ *and where the following statements hold:*

1. *$\{s_t | t = 0, 1, \ldots\}$ is a stationary, ergodic Markov process taking values in $\mathbb{R}^{2d}$*

2. *For any positive scalar $q$, there exists a scalar $\mu_q$ such that $\mathbb{E}\left[1 + \|s_t\|^q | s \equiv s_0\right] \leq \mu_q\left(1 + \|s\|^q\right)$*

3. *The step size sequence satisfies the Robbins-Monro conditions, that is $\sum_{t=0}^{\infty} \alpha_t = \infty$ and $\sum_{t=0}^{\infty} \alpha_t^2 < \infty$*

4. *There exists scalars $d$ and $q$ such that $\|\Xi(w, r)\| \leq d\left(1 + \|w\|^q\right)\left(1 + \|r\|\right)$*

5. *There exists scalars $d$ and $q$ such that $\sum_{t=0}^{\infty} \|\mathbb{E}\left[\Xi(w_t, r) | z_0 \equiv z\right] - \mathbb{E}\left[\Xi(w_0, r)\right]\| \leq d\left(1 + \|w\|^q\right)\left(1 + \|r\|\right)$*

6. *There exists a scalar $d > 0$ such that $\|\mathbb{E}[\Xi(w_0, r)] - \mathbb{E}[\Xi(w_0, \bar{r})]\| \leq d\|r - \bar{r}\|$*

7. *There exists scalars $d > 0$ and $q > 0$ such that $\sum_{t=0}^{\infty} \|\mathbb{E}\left[\Xi(w_t, r) | w_0 \equiv w\right] - \mathbb{E}\left[\Xi(w_0, \bar{r})\right]\| \leq c\|r - \bar{r}\|\left(1 + \|w\|^q\right)$*

8. *There exists some $r^\star \in \mathbb{R}^k$ such that $\Xi(r)(r - r^\star) < 0$ for all $r \neq r^\star$ and $\bar{s}(r^\star) = 0$.*

*Then $r_t$ converges to $r^\star$ almost surely.*

In order to apply the Theorem 7, we show that conditions 1 - 7 are satisfied.

*Proof.* Conditions 1-2 are true by assumption while condition 3 can be made true by choice of the learning rates. Therefore it remains to verify conditions 4-7 are met.

To prove 4, we observe that

$$\begin{aligned}
\|\Xi(w, r)\| &= \left\| \phi(s)\left(\mathcal{R}(s, \cdot) + \gamma \min\{(\Phi r)(s'), \hat{\mathcal{M}}(\Phi r)(s')\} - (\Phi r)(s)\right) \right\| \\
&\leq \|\phi(s)\| \left\| \mathcal{R}(s, \cdot) + \gamma\left(\|\phi(s')\| \|r\| + \hat{\mathcal{M}}\Phi(s')\right) \right\| + \|\phi(s)\| \|r\| \\
&\leq \|\phi(s)\| \left(\|\mathcal{R}(s, \cdot)\| + \gamma\|\hat{\mathcal{M}}\Phi(s')\|\right) + \|\phi(s)\| \left(\gamma \|\phi(s')\| + \|\phi(s)\|\right) \|r\|.
\end{aligned}$$

Now using the definition of $\mathcal{M}$, we readily observe that $\|\mathcal{M}\Phi(s')\| \leq \|c\|_\infty + \|\mathcal{R}\| + \gamma\|\mathcal{P}^{\sigma}_{s's_t}\Phi\| \leq \|c\|_\infty + \|\mathcal{R}\| + \gamma\|\Phi\|$ using the non-expansiveness of $P$.

Hence, we lastly deduce that

$$\begin{aligned}
\|\Xi(w, r)\| &\leq \|\phi(s)\| \left(\|\mathcal{R}(s, \cdot)\| + \gamma\|\hat{\mathcal{M}}\Phi(s')\|\right) + \|\phi(s)\| \left(\gamma \|\phi(s')\| + \|\phi(s)\|\right) \|r\| \\
&\leq \|\phi(s)\| \left(\|\mathcal{R}(s, \cdot)\| + \gamma\left(\|c\|_\infty + \|\mathcal{R}\| + |\phi|\right)\right) + \|\phi(s)\| \left(\gamma \|\phi(s')\| + \|\phi(s)\|\right) \|r\|,
\end{aligned}$$

we then easily deduce the result using the boundedness of $\phi$ and $\mathcal{R}$.

Now we observe the following Lipschitz condition on $\Xi$:

$$\|\Xi(w,r) - \Xi(w,\bar{r})\|$$

$$= \left\|\phi(s)\left(\gamma\min\{(\Phi r)(s'), \hat{\mathcal{M}}\Phi(s')\} - \gamma\min\{(\Phi\bar{r})(s'), \hat{\mathcal{M}}\Phi(s')\}\right) - ((\Phi r)(s) - \Phi\bar{r}(s))\right\|$$

$$\leq \gamma\|\phi(s)\|\left\|\min\left\{\phi'(s')r, \hat{\mathcal{M}}\Phi'(s')\right\} - \min\left\{(\phi'(s')\bar{r}), \hat{\mathcal{M}}\Phi'(s')\right\}\right\| + \|\phi(s)\|\|\phi'(s)r - \phi(s)\bar{r}\|$$

$$\leq \gamma\|\phi(s)\|\|\phi'(s')r - \phi'(s')\bar{r}\| + \|\phi(s)\|\|\phi'(s)r - \phi(s)\bar{r}\|$$

$$\leq \|\phi(s)\|\left(\gamma\|\phi'(s')\| + \|\phi'(s)\|\right)\|r - \bar{r}\|$$

$$\leq \text{const.}\|r - \bar{r}\|,$$

using Cauchy-Schwarz inequality and that for any scalars $a, b, c$ we have that $|\max\{a,b\} - \max\{b,c\}| \leq |a - c|$ and $|\min\{a,b\} - \min\{b,c\}| \leq |a - c|$, which proves Part 6.

Using Assumptions 3 and 4, we therefore deduce that

$$\sum_{t=0}^{\infty}\|\mathbb{E}\left[\Xi(w,r) - \Xi(w,\bar{r})|w_0 = w\right] - \mathbb{E}\left[\Xi(w_0,r) - \Xi(w_0,\bar{r})\|\right]\| \leq \text{const.}\|r - \bar{r}\|\left(1 + \|w\|^l\right).$$

$$(30)$$

which proves Part 7.

Part 2 is assured by Lemma 10 while Part 4 (and hence Part 5) is assured by Lemma 13 and lastly Part 8 is assured by Lemma 14. This result completes the proof of Theorem 4. $\qquad\square$

## PROOF OF PROPOSITION 2

*Proof.* We begin by re-expressing the *activation times* at which Switcher agent activates the Adversary. In particular, an activation time $\tau_k$ is defined recursively $\tau_k = \inf\{t > \tau_{k-1}|s_t \in A, \tau_k \in \mathcal{F}_t\}$ where $A = \{s \in \mathcal{S}, g(s_t) = 1\}$. The proof is given by deriving a contradiction. Therefore suppose that $\hat{\mathcal{M}}v_S(s_{\tau_k}) > v_S(s_{\tau_k})$ and suppose that the activation time $\tau_1' > \tau_1$ is an optimal activation time. Construct Switcher $g'$ and $\tilde{g}$ policy activation times by $(\tau_0', \tau_1', \dots,)$ and $g'^2$ policy by $(\tau_0', \tau_1, \dots)$ respectively. Define by $l = \inf\{t > 0; \hat{\mathcal{M}}\psi(s_t = \psi(s_t)\}$ and $m = \sup\{t; t < \tau_1'\}$. By construction we have that

$$v_S^{\boldsymbol{\pi},g'}(s)$$

$$= \mathbb{E}\left[\mathcal{R}_S(s_0, \boldsymbol{a}_0) + \mathbb{E}\left[\dots + \gamma^{l-1}\mathbb{E}\left[\mathcal{R}_S(s_{\tau_1-1}, \boldsymbol{a}_{\tau_1-1}) + \dots + \gamma^{m-l-1}\mathbb{E}\left[\mathcal{R}_S(s_{\tau_1'-1}, \boldsymbol{a}_{\tau_1'-1}) + \gamma\hat{\mathcal{M}}v_S^{\boldsymbol{\pi},g'}(s')\right]\right]\right]\right]$$

$$< \mathbb{E}\left[\mathcal{R}_S(s_0, \boldsymbol{a}_0) + \mathbb{E}\left[\dots + \gamma^{l-1}\mathbb{E}\left[\mathcal{R}_S(s_{\tau_1-1}, \boldsymbol{a}_{\tau_1-1}) + \gamma\hat{\mathcal{M}}v_S^{\boldsymbol{\pi},g'}(s_{\tau_1})\right]\right]\right]$$

We now use the following observation $\mathbb{E}\left[\mathcal{R}_S(s_{\tau_1-1}, \boldsymbol{a}_{\tau_1-1}) + \gamma\hat{\mathcal{M}}v_S^{\boldsymbol{\pi},g'}(s_{\tau_1})\right]$

$$\geq \min\left\{\hat{\mathcal{M}}v_S^{\boldsymbol{\pi},g'}(s_{\tau_1}), \max_{\boldsymbol{a}_{\tau_1}\in\mathcal{A}}\left[\mathcal{R}_S(s_{\tau_1}, \boldsymbol{a}_{\tau_1}) + \gamma\sum_{s'\in\mathcal{S}}P(s'; \boldsymbol{a}_{\tau_1}, s_{\tau_1})v_S^{\boldsymbol{\pi},g}(s')\right]\right\}.$$

Using this we deduce that

$$v_S^{\boldsymbol{\pi},g'}(s > \mathbb{E}\left[\mathcal{R}_S(s_0, \boldsymbol{a}_0) + \mathbb{E}\left[\dots\right.\right.$$

$$+ \gamma^{l-1}\mathbb{E}\left[\mathcal{R}_S(s_{\tau_1-1}, \boldsymbol{a}_{\tau_1-1}) + \gamma\max\left\{\mathcal{M}^{\boldsymbol{\pi},\tilde{g}}v_S^{\boldsymbol{\pi},g'}(s_{\tau_1}), \max_{a_{\tau_1}\in\mathcal{A}}\left[\mathcal{R}_S(s_{\tau_k}, \boldsymbol{a}_{\tau_k}) + \gamma\sum_{s'\in\mathcal{S}}P(s'; \boldsymbol{a}_{\tau_1}, s_{\tau_1})v_S^{\boldsymbol{\pi},g}(s')\right]\right\}\right]\right]$$

$$= \mathbb{E}\left[\mathcal{R}_S(s_0, \boldsymbol{a}_0) + \mathbb{E}\left[\dots + \gamma^{l-1}\mathbb{E}\left[\mathcal{R}_S(s_{\tau_1-1}, \boldsymbol{a}_{\tau_1-1}) + \gamma\left[Tv_S^{\boldsymbol{\pi},\tilde{g}}\right](s_{\tau_1})\right]\right]\right] = v_S^{\boldsymbol{\pi},\tilde{g}}(s)$$

where the first inequality is true by assumption on $\hat{\mathcal{M}}$. This is a contradiction since $g'$ is an optimal policy for Switcher. Using analogous reasoning, we deduce the same result for $\tau_k' < \tau_k$ after which deduce the result. Moreover, by invoking the same reasoning, we can conclude that it must be the case that $(\tau_0, \tau_1, \dots, \tau_{k-1}, \tau_k, \tau_{k+1}, \dots,)$ are the optimal activation times.

$$\square$$

# F    PROOF OF THEOREM 5

*Proof.* The proof of the Theorem is straightforward since by Theorem 4, Switcher's problem can be solved using a dynamic programming principle. The proof immediately by application of Theorem 2 in (Sootla et al., 2022).

$\square$

