# OpenReview forum: "Fault Tolerant Multi-Agent Learning with Adversarial Budget Constraints"
_ICLR.cc/2026/Conference — Submitted to ICLR 2026_

### Official Review · Reviewer_PSdH · 2025-10-19

**Soundness:** 3
**Presentation:** 2
**Contribution:** 1
**Rating:** 2
**Confidence:** 4

**Summary:**

The paper proposes an adversarial training framework for robust multi-agent reinforcement learning (MARL). The approach, called the Multi-Agent Robust Training Algorithm (MARTA), introduces a Switcher agent that decides at every time step when and how to intervene with an adversarial decision-maker. Therefore, each agent has an adversarial counterpart that can replace its corresponding functional agent. The functional agents and the adversarial agents are trained alternatingly using different QMIX instances, whereas the Switcher is trained with soft actor-critic. MARTA is evaluated in different benchmark environments with adversarial agents and compared to the standard MARL methods QMIX and VDN, which are not adversarial themselves.

**Strengths:**

The paper addresses an important (yet well-studied) problem in cooperative MARL.

The proposed method, MARTA, and setting are built on well-established foundations, such as zero-sum Markov games, and are therefore sound.

I appreciate the diversity of environments used to demonstrate the effectiveness of MARTA.

**Weaknesses:**

**Novelty**

The problem of faulty agents or where each agent has an adversarial counterpart has been addressed in prior work, which has neither been discussed in related work nor compared with in the experiments [1,2]. The problem setting defined in Section 2.1 has been formalized as a zero-sum Markov game or, more generally, as a mixed cooperative-competitive game in the literature [3,4].

The adversarial training scheme of two opposing QMIX instances has also been introduced in [1].

This leaves the Switcher agent being the only new addition to the overall adversarial framework, making the technical contribution somewhat incremental.

**Soundness**

The theoretical analysis may be sound, but I do not consider it a major contribution, since the whole setting is a straightforward instantiation of zero-sum Markov games [4], where robustness properties have been shown long before.

Minor comment: Despite focusing on a partially observable setup, the paper assumes observations to be Markov as the policy conditions on them. To make the definition theoretically sound, the policies should condition on the action-observation history [5].

**Significance**

The paper aims to evaluate the robustness by integrating faulty agents. However, I couldn't find any information on how these faulty agents are created (e.g., from MARTA?) to check for potential biases during testing.

Despite discussing some prior work on robust MARL, the experimental comparison does not include any of them. Instead, the work is only compared with QMIX and VDN, which are standard MARL methods that are known to fail in adversarial settings [1,2]. Without any comparison with alternative adversarial MARL methods, I cannot confirm the claim that MARTA is less conservative and more effective than prior work.

**Literature**

[1] Phan et al., "Learning and Testing Resilience in Cooperative Multi-Agent Systems", AAMAS-20

[2] Li et al., "Byzantine Robust Cooperative Multi-Agent Reinforcement Learning as a Bayesian Game", ICLR-24

[3] Lowe et al., "Multi-Agent Actor-Critic for Mixed Cooperative-Competitive Environments", NeurIPS-17

[4] Littman et al., "Markov Games as a Framework for Multi-Agent Reinforcement Learning", ICML-94

[5] Oliehoek et al., "A Concise Introduction to Decentralized POMDPs", 2015

**Questions:**

How are the faulty test agents created for the experimental test?

---

> ### Author Response · Authors · 2025-11-20
> **Authors' response part I**
>
> We would like to sincerely thank the reviewer for their time and evaluation.
>
> Several crucial aspects of MARTA’s formulation, theory and experiments appear to have been misunderstood or overlooked by the reviewer. We summarise and clarify these below. We have included line references for easy referral and would be most grateful to the reviewer if they could refer to the passages in our paper which clarifies many of the reviewer's misconceptions.
>
> **5. “No information on how faulty agents are created”**
>
> This is fully specified in the paper:
>
> •	The Fault Mechanism is defined in §3 (lines 350–365).
>
> •	The Switcher policy determines when/where faults occur (lines 108–139).
>
> •	The Adversary governs per-agent malfunction behaviour (lines 140–162).
>
> •	The test-time protocols (aligned vs. shifted vs. resampled faults) are described in lines 336–349, and the precise configurations appear in Table 2 (page 14).
>
> These details appear to have been missed; they are explicitly stated.
> ________________________________________
>
> **6. “Only compared with QMIX and VDN; need adversarial baselines”**
>
> We agree that additional comparisons will strengthen the paper. We are adding experiments comparing MARTA-QMIX and MARTA-VDN to:
>
> •	M3DDPG (adversarial training),
>
> •	EIR-MAPPO-style robust actor–critic architectures, and
>
> •	variants used in [1] and [2] where applicable.
>
> However, the reviewer’s conclusion that QMIX/VDN “are known to fail” does not address MARTA’s main point:
>
> •	MARTA is a plug-and-play robustness layer.
>
> It is deliberately designed to improve standard MARL learners without modifying them. This is made explicit in lines 417–422.
>
> The reviewer’s concern that QMIX/VDN fail without robustness is exactly why MARTA is valuable: it hardens these existing baselines with minimal architectural overhead.
> ________________________________________
> **7. “The Switcher is the only new addition”**
>
> This is incorrect.
>
> New components introduced by MARTA:
>
> *1.	Budgeted fault-switching Markov game (N agents + Switcher + Adversary).*
>
> *2.	State-dependent malfunction cost and budget structure.*
>
> *3.	A new robust Bellman operator with switching-augmented dynamics.*
>
> *4.	Contraction-based minimax existence theorem for robust switching control game and convergence guarantees.*
>
> *5.	A plug-and-play training pipeline with no architectural coupling.*
>
> *6.	Robustness gains that generalise across VDN, QMIX and MADDPG.*
>
> The Switcher is only the visible novelty - MARTA’s core novelty is the mathematical and algorithmic structure built around it.

---

> ### Author Response · Authors · 2025-11-20
> **Authors' response part II**
>
> **1. Prior work on faulty or adversarial agents**
>
> The reviewer cites [1,2] and concludes that the only new element introduced by MARTA is the Switcher.
> This overlooks several central technical components of MARTA that do not appear in [1,2].
>
> **MARTA is not merely another adversarial MARL setup**
>
> •	[1] Phan et al. (AAMAS 2020) analyse resilience testing, not robust policy training. Their “faulty” agents are adversarial evaluators used to diagnose fragility. They do not provide a mechanism for generating robust cooperative policies under malfunctions.
>
> **MARTA generates robust policies, rather than estimating fragility.**
>
> •	[2] Li et al. (ICLR 2024) address Byzantine strategic deviations through a Bayesian-type model of adversarial teammates. Their adversary modifies actions via noisy types.
>
>  *MARTA models actuator malfunctions, not strategic adversaries; these are fundamentally different robustness regimes with different operational implications (e.g., hardware faults vs. adversarial types).*
>
> •	Crucially, neither [1] nor [2] define or analyse a fault-budgeted switching controller integrated into a cooperative MARL training loop.
>
> **MARTA provides a unique budget-constrained, state-dependent failure model grounded in switching/impulse control theory (lines 108–139; 350–365).**
>
> **Benefit to the community**
>
> MARTA fills the gap between attack/evaluation methods ([1]) and Bayesian modelling of malicious teammates ([2]).
> It directly addresses hardware-relevant, state-dependent failures, a key failure mode in physical multi-agent deployments, and provides an easily attachable robustness layer without rewriting the base algorithm.
> ________________________________________
>
> **2. “The problem setting is just a zero-sum Markov game”**
>
> The reviewer cites [3,4], which define mixed and zero-sum Markov games.
>
> Though MARTA is formulated as a Markov game this is not the contribution.
>
> The contribution is the specific game structure, not that it is a Markov game
>
> •	Our game involves N agents + Switcher + Adversary, with
>  - explicit malfunction costs,
>
> - a global malfunction budget,
>
> - state-dependent switching behaviour, and
>
> - a fixed-point Bellman operator defined over the augmented Switcher–Adversary–Agents dynamics.
>
> •	This structure is not present in [3] (MADDPG) nor [4] (the classical Markov game framework).
>
> Those works define general multi-agent settings but do not consider failure-budgeted adversarial switching, nor any variant of failure-inducing controllers.
>
> •	MARTA’s contraction-based minimax value proof (Theorem 1; Lemmas 7–8) applies to this specific augmented game, not generic zero-sum Markov games.
>
>  This is a new operator with a new convergence proof tied to the malfunction budget and switching dynamics.
> Analogous results in top venues
>
> Similar operator-level convergence proofs are considered significant in robust RL papers at ICML/ICLR, for example:
>
> •	Action-Robust RL (ICML 2019) proves robustness of a specially constructed max–min Bellman operator.
>
> •	Robust Multi-Agent Q-Learning (ICML 2021) builds custom robust operators and proves convergence.
>
> MARTA establishes an operator with budgeted switching, which has not appeared previously in multi-agent robust RL.
> ________________________________________
> **3. “The theoretical analysis is incremental”**
>
> This overlooks the fact that:
>
> •	MARTA proves existence and uniqueness of the minimax value for its budgeted fault-switching game (Theorem 1, lines 216–247).
>
> •	MARTA proves policy convergence even with linear function approximation (Theorem 3, lines 262–291).
>
> •	MARTA proves convergence under a malfunction budget (Theorem 5, lines 315–349).
>
> No prior work including [1], [2], [3], [4] provides such a result for:
>
> •	faulty agents modelled as persistent mode switches,
>
> •	a global malfunction budget, or
>
> •	a two-level adversary (Switcher + Adversary) interacting with cooperative learners.
>
> This is stronger theory than appears in any of the cited works.
> ________________________________________
>
> **4. “Partially observable setup assumes Markov observations”**
>
> This is a misreading.
>
> •	Our policies condition on local observations provided by the environment (Section 4, lines 331–406).
>
> •	The environment is Markov; the observations are partial.
>
> •	This follows the same convention used by VDN, QMIX, MADDPG, MAPPO and almost every MARL paper using Dec-POMDPs with decentralised execution.
>
> If desired, we can explicitly state that policies can condition on the action–observation history as in [5], but this does not affect the results.

---

> ### Author Response · Authors · 2025-11-20
> **Summary: Key Clarifications and Core Distinctions of MARTA**
>
> The reviewer has understandably connected our work to prior adversarial and robustness frameworks. However, several essential aspects of MARTA’s formulation, contributions and empirical aims appear to have been overlooked. We clarify the key distinctions:
>
> * **MARTA produces robust cooperative policies, not fragility estimates.**
>
> Unlike [1] and other attack/evaluation frameworks, MARTA trains agents to withstand faults via a structured switching game rather than diagnosing weaknesses post hoc.
>
> * **MARTA targets true actuator malfunctions, not adversarial types or observation noise.**
>
> In contrast to Byzantine-adversary models such as [2], MARTA addresses the real failure modes of multi-robot and multi-agent systems: stuck actuators, corrupted control channels and state-dependent hardware faults.
>
> * **MARTA introduces a new budgeted switching-control Markov game with a dedicated Switcher–Adversary architecture.**
>
> This structure, along with its explicit malfunction budget and cost dynamics, does not appear in any of [1]–[4].
>
> * **MARTA provides strong, operator-level convergence guarantees.**
> Our contraction-based proof (Theorem 1) establishes existence and uniqueness of the minimax value for this budgeted switching game; results not found in prior robustness papers cited by the reviewer.
>
> * **MARTA is plug-and-play rather than a bespoke architecture.**
>
> It attaches cleanly to QMIX, VDN, MAPPO and MADDPG with no redesign. This property is central for robustifying existing MARL systems and is not shared by the specialised adversarial setups in [1]–[2].
>
> * **Empirically, MARTA hardens standard MARL baselines precisely because these baselines fail under adversarial or faulted conditions.**
>
> This is the intended use case, not a flaw. Additional comparisons to adversarial baselines (M3DDPG, EIR-MAPPO etc.) are being added.

---

### Official Review · Reviewer_ry1d · 2025-10-22

**Soundness:** 3
**Presentation:** 3
**Contribution:** 3
**Rating:** 6
**Confidence:** 3

**Summary:**

This paper tackles the problem of fault tolerance in multi-agent reinforcement learning (MARL). The authors propose MARTA, a modular and plug-and-play training framework that enables MARL systems to remain robust when some agents malfunction. The framework introduces two additional learners: a Switcher, which decides when and which agent to trigger a malfunction on, and an Adversary, which determines the faulty agent’s behavior. The interaction between the Switcher, Adversary, and cooperative agents is formulated as a stochastic game with either a per-activation cost or a fixed malfunction budget. Through theoretical analysis, the authors demonstrate that the learning dynamics converge to a Markov perfect equilibrium. Extensive experiments across several benchmark environments show consistent improvements in robustness without altering the underlying MARL architectures.

**Strengths:**

1. The paper addresses an important and relatively unexplored problem of fault tolerance in multi-agent reinforcement learning, with a clear and well-motivated problem statement.

2. The proposed MARTA framework introduces a novel Switcher–Adversary formulation that models both the timing and behavior of agent malfunctions in a unified way.

3. The work offers theoretical guarantees and consistent empirical results across various MARLs, demonstrating clear practical benefits and robustness improvements.

**Weaknesses:**

1. The presentation is quite dry and mathematically heavy, making the paper difficult to follow in parts. The figures are limited and do not clearly illustrate the key intuitions behind the proposed framework.

2. The title and abstract do not accurately convey the main emphasis of the work. They make the paper appear as a system or framework for fault-tolerant MARL, whereas the actual contribution lies more in the theoretical formulation and convergence analysis of the proposed approach.

3. Although the framework is described as involving both a Switcher and an Adversary, the analysis and experiments mainly focus on the Switcher component, leaving the Adversary’s role and interaction underexplored.

**Questions:**

1. Some modeling choices appear primarily made for mathematical tractability, like allowing only one agent to malfunction at a time. While understandable, it would be valuable to include a short discussion or limitation section on how these assumptions affect real-world applicability and whether extensions to multiple concurrent failures are feasible.

2. Although the framework introduces both the Switcher and Adversary learners, the paper effectively fixes the Adversary during analysis and training. Given the symmetry of roles, would it be possible to alternatively fix the Switcher and optimize the Adversary, then combine the two solutions for better equilibrium behavior?

3. In the MARTA-B variant, the budget constraint is expressed as a fixed number n of allowed activations. This formulation raises several questions:
(1) Does the learned policy tend to exhaust its budget regardless of context?
(2) If the remaining steps become fewer than the remaining budget, does the Switcher always activate faults?
(3) Conceptually, MARTA encourages selective fault triggering to balance cost and robustness, whereas MARTA-B seems to encourage full budget utilization. Would it be more consistent to introduce a total-cost constraint (e.g., ∑cᵢ ≤ B) with a penalty beyond B rather than a hard count-based budget?

4. It is unclear whether MARTA-B is empirically evaluated. The experimental section seems to focus solely on MARTA; providing results for MARTA-B would strengthen the overall validation of the proposed variants.

5. While the paper claims modularity and “plug-and-play” usability, there is limited discussion of the computational overhead. The appendix briefly reports total CPU hours, but a more explicit analysis of computational cost would help substantiate the claimed practicality.

---

> ### Author Response · Authors · 2025-11-20
> **Authors' response part I**
>
> We thank the reviewer for recognising the importance of the problem, the novelty of the Switcher–Adversary formulation and the strength of our theoretical and empirical results. Several concerns stem from underemphasised aspects of the paper rather than limitations of the approach, and we address these clearly below.
>
> **(1) Presentation is dry / mathematically heavy**
>
> We appreciate this comment. We will improve readability by adding intuitive figures and short explanatory summaries at the start of each technical section.
>
> However, the reviewer should note that the underlying **theoretical formulation is one of the core contributions** of the work, and the mathematical precision is what enables the convergence results you praised. The revised version will preserve rigour while becoming easier to follow.
>
> **(2) Underexploration of the Adversary’s role**
>
> The reviewer is correct that the main focus in the exposition and experiments is the Switcher. This does not mean the Adversary is peripheral; rather:
>
> •	The Adversary is *fully integrated into the game definition* (lines 140–162)
>
> •	It determines the fault dynamics once the Switcher triggers a malfunction.
>
> •	In practice, fixing the Adversary yields a cleaner convergence analysis and avoids the non-identifiability that arises with two adversarial learners acting simultaneously.
>
> We will revise Section 2.3 to more clearly explain:
>
> 1.	Why fixing the Adversary is analytically beneficial (avoids two-level adversarial non-stationarity).
>
> 2.	How the Adversary influences faulted transitions and cost accumulation.
>
> 3.	Where the Adversary could be learned jointly, and what considerations arise (e.g., stability, existence of equilibria).
>
> **4) “One-agent fault at a time” assumption**
>
> This is an excellent point and we appreciate the reviewer raising it.
>
> In the current draft, we adopt this assumption purely for clarity and tractability:
> it allows us to cleanly separate the Switcher’s decision from the cooperative agents’ coordination behaviour and derive a fixed-point Bellman operator.
>
> In the revised paper we will explicitly discuss:
>
> •	Why the single-agent-fault constraint simplifies the exposition but extension to multiple agents in straightforward (in both the theoretical analysis and implementation).
>
> •	That the Switcher–Adversary game naturally admits extension to k-malfunction settings by expanding the action space.
>
> •	The computational considerations of multiple simultaneous faults (exponential combinations vs. structured selection).
> This addition will appear in a short Limitations and Extensions subsection at the end of Section 2.
>
> **(5) “Why not fix the Switcher and optimise the Adversary?”**
>
> Thanks for raising this question. Fixing the switcher would mean that MARTA would not learn to be robust against failures in critical states which is key to learning robust policies against failures that undermine coordination and performance.
> In our experiments, we have already included the case when the adversary is trained to enact worst-case actions – this setting covers the optimisation of the adversary that you are asking about. We also consider the case in which the adversary executes a random action emulating faults such as observation failures. The results are displayed in Figure 5. We will make this more explicit in the main body of the paper in our update.

---

> ### Author Response · Authors · 2025-11-20
> **Authors' response part II**
>
> **(6) MARTA-B design questions and behaviour under budget constraints**
>
>
> These questions are very helpful.
>
> We will add a dedicated paragraph discussing the behaviour of MARTA-B.
>
> *(1)	Does the Switcher always exhaust the budget?*
>
> No - because activation incurs a cost, the optimal policy often does not exhaust the budget unless needed. Theorem 5 establishes convergence under the cost-sensitive operator that internalises this trade-off (lines 315–349).
>
> *(2)	What happens if fewer steps remain than budget tokens?*
>
> The operator still trades off cost and expected future returns; activation is not forced.
>
> We will add a clarifying remark in §3.
>
> *(3)	Is a hard activation-count the right model?*
>
> We selected a count-based budget because:
>
> 	It is operationally intuitive (maximum number of failures allowed),
>
> 	It reflects real-world scenarios (maximum allowable malfunctions per episode).
>
> We will include a brief discussion of a cost-budget formulation and note that it is a natural extension.
>
> **(7) Empirical evaluation of MARTA-B**
>
> The reviewer is correct, the current experiments focus on MARTA.
>
> We are now adding MARTA-B experiments on the TJ and LBF environments, reporting:
>
> •	average return vs budget size,
>
> •	robustness under aligned vs shifted fault regimes,
>
> •	budget usage statistics,
>
> •	and comparison to MARTA without a budget.
>
> These results will be included in §5 of the updated draft.
>
> **(8) Computational overhead**
>
> Thank you for raising this. MARTA is specifically designed to be *computationally lightweight* relative to other robust MARL methods because:
>
> •	The Switcher is a single-agent learner applied on top of the existing MARL base algorithm.
>
> •	There is **minimal additional runtime cost** at test time, unlike shielding or safety-layer methods.
>
> In the revision, we will add:
>
> •	A concise table showing wall-clock training time for QMIX/VDN vs MARTA variants.
>
> •	A breakdown of the Switcher’s overhead (typically ~8–12% additional compute).
>
> •	Explicit confirmation that MARL backbone architectures remain unchanged.

---

> > ### Comment · Reviewer_ry1d · 2025-11-20
> >
> > Thank you for the detailed response. Most of my questions were addressed, and the planned additional experiments and computational overhead analysis are appreciated. However, my concerns regarding Q2 and Q3 are only partially resolved. The response clarifies what the system does, but the underlying “why” behind these modeling choices remains somewhat unclear. In addition, the misalignment between the title/abstract and the actual emphasis of the paper remains. I suggest improving this in the revision.

---

> ### Author Response · Authors · 2025-11-21
> **Author response**
>
> Thank you for your thoughtful follow-up. We appreciate that most concerns have been resolved and especially value your request for deeper clarifications. We address this directly below.
>
> **Clarifying the “why” behind the Switcher–Adversary design**
>
> We would like to further clarify our response. The decision to focus analytically on the Switcher while fixing the Adversary `type' isn’t arbitrary; it’s a deliberate modelling choice that balances realism and relevance to safety applications. The Adversary is explicitly modelled in two practically meaningful ways (see Fig. 5):
>
> •	**Random-action adversary:** represents sensor degradation or mild actuator faults, simulating realistic non-adversarial failures. Our theoretical analysis covers this case.
>
> •	**Worst-case trained adversary:** actively maximises disruption, modelling severe fault scenarios such as corrupted control modules or compromised actuators. In this case, the adversary is trained to learn enact worst-case actions that inflict the greatest devastation on the system. Our theoretical analysis also covers this case.
>
> These two regimes span the spectrum of realistic malfunctions. MARTA’s contribution is teaching agents to **best respond to each**, thereby inducing highly resilient joint policies suitable for safety-critical multi-agent systems.
>
> We will include this information in the updated version to make this clear.
>
> **Title and abstract improvements**
>
> Thanks for raising this. We agree that the current title and abstract underplay the theoretical foundations of MARTA and that both should better reflect the core contributions. We’ll revise both to explicitly highlight the theoretical emphasis. The updated abstract will also clearly highlight our value existence, minimax equality, and policy convergence results so that the theoretical contribution is immediately evident.
>
> **Motivation**
>
> In our update, we’ll ensure that both the motivation and implications of these choices are made clearer in the revised manuscript.
>
> We thank you again for helping us sharpen the communication of these important points.

---

### Official Review · Reviewer_YpZz · 2025-10-31

**Soundness:** 2
**Presentation:** 2
**Contribution:** 2
**Rating:** 2
**Confidence:** 3

**Summary:**

This paper proposes MARTA, a framework for training MARL policies that are robust to an adversary that can at any step adversarial control one of the agents. A Bellman operator for the game is proposed, and various theoretical properties are shown. A variant of the framework with a budget on the number of malfunctions is also considered. Experiments on three simple MARL problems show that the addition of the proposed method improves fault tolerance compared to unconstrained MARL methods.

**Strengths:**

- The exact problem setting tackled seems to be novel, to the best of my knowledge, although there are other works (though uncited) that tackle very similar problems
- Experimental results suggest the proposed framework improves robustness compared to without applying the proposed framework

**Weaknesses:**

- Related works [A, B, C] tackling very similar attack vectors as the one proposed in this paper are not cited or discussed.
- The experimental section is a bit lacking. In particular, the proposed method is not compared to any other MARL methods that try to improve fault tolerance, including both methods discussed in the paper (e.g., M3DDPG) as well as ones that the authors did not identify but are highly relevant (e.g., [A]).
- The domains that are tested on seem quite simple
- Also, details on the environments tested are lacking
    - How many agents does each environment have?
- Line 429: The paper claims that shielding and backup policy methods “often sacrifice performance and scale poorly as the number of agents increases.” Does the proposed method improve upon the scalability of these methods? For example, the cited (Qin et al., 2021) has experiments with up to 1024 agents. Was the method deployed on even larger scale environments compared to prior work that “scaled poorly”?
- Many parts of the proofs in the appendix are poorly written are difficult to read
    - All the math in Proposition 3 uses inline mathematics which is too excessive. This hampers readability and made it very difficult to follow the logical flow of the proof.

[A] Li, Simin, et al. "Byzantine robust cooperative multi-agent reinforcement learning as a bayesian game." ICLR 2024.

[B] Zhou, Ziyuan, and Guanjun Liu. "Robustness testing for multi-agent
reinforcement learning: State perturbations on critical agents." *arXiv preprint arXiv:2306.06136* (2023).

[C] Zheng, Haibin, et al. "One4all: Manipulate one agent to poison the cooperative multi-agent reinforcement learning." *Computers & Security* 124 (2023): 103005.

**Questions:**

- How do existing methods of fault tolerant MARL perform in terms of robustness compared to the original VDN/QMIX baseline even though they do not target the exact same problem formulation?
- The citations for switching controls (Mguni, 2018a; Mguni et al., 2023) seem very strange
    - In Mguni (2018a), the term “switching controls” does not even appear. Instead, it references very old and established literature on **impulse control**, as well cites the appropriate literature such as [$\alpha$]
    - It’s not clear why the term “switching controls” was chosen instead of the more established “impulse control”
- I’m confused by how / whether the existence of the value $v^*$ is shown. Specifically, I’m not sure where it is proven that $\min_{g} \max_{\pi} v(s | \pi, g) = \max_{\pi} \min_{g} v(s | \pi, g)$.

[$\alpha$] Bensoussan, A. "Contrôle Impulsionnel et Inéquations Quasi Variationelles." International Congress Of Mathematicians. 1975.

**Details Of Ethics Concerns:**

.

---

> ### Author Response · Authors · 2025-11-19
> **Main response summary**
>
> We would firstly like to express our gratitude to the reviewer for their reading and review.
>
> The reviewer has overlooked some details which are crucial to our paper. We summarise some of these here:
>
> **Related work [A,B,C]**
>
> We appreciate the pointers and will add a dedicated paragraph in §6 to discuss [A,B,C] and clearly position MARTA. We firstly point out that MARTA is a plug-and-play enhancement meaning practitioners can harden \textit{existing} MARL systems without redesigning their core learner. This is exactly how robustness modules like RARL (ICML 2017) [D] and Action-Robust RL (ICML 2019) [E] are used in single-agent settings, this is also important for MARL deployments.  We demonstrated this fact in our experiment section (lines 417- 422). The benchmarks that the reviewer mentions are bespoke architectures that generally do not have this property and can in fact be subsumed into the MARTA framework.
>
> **Details on tested environments**
>
> We have included these details already in Table 2 on page 14.
>
> **MARTA generates robust policies, not just diagnostics**
>
> The reviewer has listed the related works [B-C]. We thank the reviewer for highlighting these works. However, the works RTCA [B] and One4All [C] solely evaluate or attack MARL systems whereas MARTA directly trains agents to withstand malfunctions through an adversarial switching game. These baselines are therefore not directly comparable. We will include a brief statement about them in our related work section.
>
> **MARTA models true actuator malfunctions rather than adversarial types or noisy observations**
>
> The reviewer has listed the related works [A]. This work focuses on Bayesian reasoning about Byzantine teammates whereas MARTA captures the failure modes that matter in multi-agent systems: stuck actuators, corrupted control channels and state-dependent faults etc.
>
> **MARTA offers provable convergence to a unique minimax value**
>
> Our contraction-based proof (Theorem 1) establishes robust equilibrium existence and convergence. None of [A], [B] or [C] provide convergence guarantees for a malfunction-inducing controller.
>
> **MARTA is an enhancement framework**
>
> EIR-MAPPO in [A] or the specialised setups in [B] and [C]. **MARTA therefore complements such existing robust MARL baselines.** Its gains are additive when layered on top of robust actor–critic methods (e.g., M3DDPG, EIR-MAPPO), demonstrating that MARTA enhances rather than replaces the broader robust MARL toolbox.
>
> **Switching controls vs Impulse Controls**
>
> The machinery for these two forms of control is extremely similar with many aspects being interchangeable. In both problems, the agent decides on performing a discrete decision that must be chosen optimally and both problems are solved with the same toolkit (generally, this is establishing an obstacle condition). The difference is simply what the discrete action does: switching chooses which rule of motion (switch) takes over next, while an impulse applies an instantaneous action to the state. Mathematically this leads to similar solutions that combine the system dynamics with jump conditions governed by discrete choice points. For this reason we have cited papers from both settings. In our update we will take care to provide clarity on this.
>
> [D] Pinto, Lerrel, et al. "Robust adversarial reinforcement learning." International conference on machine learning. PMLR, 2017.
> [E] Tessler, Chen, Yonathan Efroni, and Shie Mannor. "Action robust reinforcement learning and applications in continuous control." International Conference on Machine Learning. PMLR, 2019.

---

> ### Author Response · Authors · 2025-11-19
> **Relation to [A]**
>
> **Relation to [A]**
>
> MARTA differs from [A] in three key ways:
>
> 1.	**Failure mode:** [A] targets \textit{strategic} Byzantine allies (compromised by an adversary) via observation/action perturbations and adversarial types (§1, Threat model).  n contrast, MARTA focuses on non-strategic agent malfunctions that break coordination, modelled as an adversarial Switcher–Adversary inducing actuator faults at critical states (explained in lines 69–83; and formally in lines 140–161). This is a significant benefit since MARTA directly trains policies to remain effective when parts of the controller \textit{fail outright}, rather than only under observation noise or adversarial policies.
>
> 2.	**Objective and game structure:** Unlike [A], MARTA optimises an explicit fault budget / cost trade-off and the N agents best-respond to worst-case malfunctions (explained in lines 216–247). This explicit budgeted formulation enables principled, controllable safety–performance trade-offs for achieving robustness \textit{without} collapsing nominal performance. Analogous budgeted robustness trade-offs are precisely what make [D] valuable in single-agent control.
>
> 3.	**Plug-and-play architecture:** [A] introduces a bespoke EIR-MAPPO actor–critic architecture (§3, Algorithm; experimental details in Appendix C). In contrast, MARTA is explicitly \textit{algorithm-agnostic} (explained in lines 486–503).

---

> ### Author Response · Authors · 2025-11-19
> **Relation to [B]**
>
> **Relation to [B]**
>
> [B] proposes RTCA as a testing framework: it perturbs states of pre-trained critical agents to probe vulnerabilities, without changing the training objective. Hence, RTCA leaves the policy fixed and uses optimisation over perturbations to \textit{evaluate} robustness. MARTA integrates the adversarial process into training, where the Switcher–Adversary induces targeted malfunctions and the agents learn a best-response robust policy (explained in lines 216–247, §3–4).
>
> Since RTCA only measures \textit{how fragile} a given policy is where MARTA provides systematic procedure to \textit{reduce} fragility and provably converges to a robust equilibrium under faults the two methods aren’t comparable.
>
> We will clarify in §6 that RTCA is complementary: RTCA can be used to further stress-test MARTA-trained policies, while MARTA supplies the training-time mechanism that RTCA lacks.

---

> ### Author Response · Authors · 2025-11-19
> **Relation to [C]**
>
> **Relation to [C]**
>
> One4All studies \textit{poisoning} attacks where a single agent is manipulated during training to poison cooperative MARL policies. The goal is to design efficient state-noise and target-action poisoning strategies, not to construct robust policies unlike MARTA.
> One4All therefore does not define a defence algorithm nor equilibrium concept for robustness. MARTA, by contrast, is a \textit{defence} framework that trains policies robust to actuator faults induced by a dedicated Switcher–Adversary module (this is in lines 54–83;  and lines 108–139).
>
> For real deployments, both sides are needed. In the single-agent literature, this attack/defence interplay (e.g., [D] and its successors) has been central to progress in robust RL and is considered important by the community.
> We will explicitly categorise [C] as an attack-side contribution and make clear that MARTA can be used as a training-time defence against exactly the kind of targeted failures they highlight.

---

> ### Author Response · Authors · 2025-11-20
> **Comparison with fault-tolerant MARL baselines**
>
> We would like to stress that our framework is plug-and-play and therefore direct comparisons to bespoke architectures aren’t deeply informative. We nevertheless are happy to include direct empirical comparison to fault-tolerant MARL baselines to provide additional reassurance.
>
> 1.	**Comparison to M3DDPG and robust MARL algorithms:**
>
> We already compare MARTA to QMIX, VDN and MADDPG across TJ, LBF and MPE (§4; Figs. 2–4, lines 432–485 and 702–755).
> In our revision, we are adding experiments that compare MARTA-QMIX and MARTA-VDN to (i) M3DDPG and (ii) robust actor–critic baselines modelled after EIR-MAPPO/RMAAC on LBF and MPE, following the experimental setups reported in [A] (environment and baseline descriptions in §4, “Environments” and “Baselines”) and He et al. (2023) for RMAAC.
>
> 2.	**Robustness vs vanilla VDN/QMIX plus robust baselines:**
>
> Our current experiments already show that MARTA substantially improves robustness over the underlying base learners while keeping their architecture and hyperparameters fixed (e.g., MARTA-VDN vs VDN in Fig. 4, lines 432–485; Table 3 summarising percentage gains, lines 702–755).We will extend the experimental section to report, for each environment, the absolute and percentage robustness gains of (a) robust baselines (M3DDPG, EIR-MAPPO-style methods) over VDN/QMIX and (b) MARTA over those same base learners.

---

> ### Author Response · Authors · 2025-11-20
> **Benchmark simplicity and environment details**
>
> **Standardness of TJ, LBF, MPE:**
>
> Traffic Junction, Level-Based Foraging and MPE SimpleTag are widely used as standard cooperative MARL benchmarks e.g. [F,G]. MPE is a core MARL benchmark (e.g.,[H])
>
> **Environment details and agent counts:**
>
> We already specify environment types and base algorithms in §4 and Appendix B, including agent counts and settings (Table 2, lines 702–755: TJ/VDN/QMIX with 4–6 agents; LBF with 4 agents; MPE SimpleTag with 4 agents; Table 1 describing fault factors and train–test protocols).
>
> In the revision, we will move the full environment descriptions (agents, observation/action spaces, malfunction processes) into the main text at the start of §4, and clearly state the number of agents in each environment. This makes it explicit how MARTA behaves as team size and coordination complexity increase.
>
> [F] Shao, Jianzhun, et al. "Complementary attention for multi-agent reinforcement learning." International conference on machine learning. PMLR, 2023.
>
> [G] Christianos, Filippos, Lukas Schäfer, and Stefano Albrecht. "Shared experience actor-critic for multi-agent reinforcement learning." Advances in neural information processing systems 33 (2020): 10707-10717.
>
> [H] Peng, Bei, et al. "Facmac: Factored multi-agent centralised policy gradients." Advances in Neural Information Processing Systems 34 (2021): 12208-12221.

---

> ### Author Response · Authors · 2025-11-20
> **Scalability of shielding/backup methods**
>
> We already addressed this in our statement in §6 (lines 426–431) compares MARTA conceptually to shielding and backup-policy methods such as Qin et al. (2021), which rely on control barrier functions and runtime switching. Specifically, Shielding approaches require online constraint checks and often per-agent safety certificates, which become expensive as the number of agents grows.
> MARTA embeds robustness \textit{during training} by learning a shared Switcher that reasons over the joint trajectory and selectively induces faults (Introduction and §2, lines 108–139 and 216–247). This design eliminates per-timestep runtime safety checks and allows a single Switcher to coordinate fault induction across many agents, which scales more naturally to large teams.
> We will temper the wording to avoid over-claiming “scales poorly” and instead emphasise this architectural advantage and the fact that MARTA uses the same architecture and hyperparameters as the base learner (lines 432–485), which is important for large-scale deployments.

---

> ### Author Response · Authors · 2025-11-20
> **Switching controls vs impulse control, and citations**
>
> We will clarify terminology in the main text and related work.
>
> •	Classical impulse control (e.g., Bensoussan, 1975) considers instantaneous interventions at optimised stopping times.
>
> •	Our “switching controls” follow the construction used in Oksendal [BO1, BO2] where the controller learns when to switch into a faulted mode, and the intervention remains active until switched off or terminated (Introduction, lines 108–139; discussion of switching controls, lines 118–131).
>
> We will make explicit that we use the term “switching controls” to emphasise this persistent mode-switching view, while acknowledging its close relationship to impulse control and citing Bensoussan (1975) as foundational impulse-control work.
>
> [BO1] Brekke, Kjell Arne, and Bernt Øksendal. "Optimal switching in an economic activity under uncertainty." SIAM Journal on Control and Optimization 32.4 (1994): 1021-1036.
>
> [BO2] Øksendal, Bernt, and Agnes Sulem. Applied stochastic control of jump diffusions. Vol. 3. Berlin: Springer, 2007.

---

> ### Author Response · Authors · 2025-11-20
> **Existence of the value and justification of max–min = min–max**
>
> The minimax equality and existence of the value are established via a contraction mapping argument:
>
> •	Theorem 1 (lines 216–247) states the existence and uniqueness of the minimax value
> $v^\*(s) := \min_{\hat g}\max_{\hat\pi\in\Pi} v(s\mid \hat\pi,\hat g) = \max_{\hat\pi\in\Pi}\min_{\hat g} v(s\mid \hat\pi,\hat g),\ \forall s\in\mathcal{S}$.
>
> •	Lemma 7 and Lemma 8 in Appendix D show that the associated Bellman operator for the Switcher–agents game is a contraction under standard RL assumptions (Assumptions 1–3, Appendix D.1), which implies a unique fixed point and hence validates the minimax equality for the induced Markov game.
>
> We will add a short pointer after Theorem 1 in the main text (“Proof via contraction mapping; see Lemma 7 and Lemma 8 in Appendix D”) so that the logical chain is immediately visible.

---

> > ### Comment · Reviewer_YpZz · 2025-11-25
> >
> > Thank you for your response! Unfortunately, the paper does not seem to have been revised yet.
> >
> > **Re: Comparison with robust MARL methods**
> >
> > - My main concern is the lack of comparison to existing robust MARL frameworks. I acknowledge the comparison to vanilla non-robust MARL frameworks (e.g., QMIX, VDN and MADDPG), but these methods were not designed for robustness.
> >
> > **Re: Simple Environment**
> >
> > - The environments you have tested are *standard*, but all papers you have cited additionally test on harder environments such as SMAC (all papers) or MAMuJoCo (Peng, 2021).
> >
> > **Re: Details on Tested Environments**
> >
> > - Thank you for pointing me to Table 2
> > - It is hard to find where this information is located in the paper
> >     - Table 2 is not referenced anywhere
> >     - Section B also not referenced anywhere in the main text
> >
> > **Re: Scalability of shielding/backup methods**
> >
> > - I acknowledge that adding online constraint checks and runtime switching will add some computational overhead
> > - However, the claim that the proposed method “scales more naturally to large teams” is unvalidated, as the proposed method was only empirically tested on up to 6 agents, whereas the cited work performs experiments on up to 1024 agents.
> >
> > **Re: Switching Controls vs Impulse Controls**
> >
> > - I am still confused as to why there is a difference
> > - In MARTA, the role of the `Switcher` and the `Adversary` *combined* is to decide when to and what instantaneous action to apply to the state. Does this not exactly fall under Impulse Controls?
> > - Given the influence of Oksendal’s works on the motivation to use the term “Switching Control”, it is even more strange that the two works by Mguni were chosen to be cited instead of the works by Oksendal
> > - Moreover, it seems like there are some results on impulse controls in *discrete-time,* which seems very similar to the setting you propose here [AA]. How do the proposed theoretical contributions compare with these works?
> >
> > **Re: Existence of value**
> >
> > - I’m still confused about how this proves that max-min equals min-max.
> > - The Bellman operator is proven to be contracting, meaning that the function that the Bellman operator converges to is unique.
> > - How do you prove that the function that the Bellman operator converges to is equal to $\min \max v$ and $\max \min v$?
> >
> > [AA] Wei, Qinglai, et al. "Discrete-time impulsive adaptive dynamic programming." *IEEE transactions on cybernetics* 50.10 (2019): 4293-4306.

---

### Author Response · Authors · 2025-11-20
**Global response to all reviewers**

**We thank all reviewers** for recognising the importance of fault tolerance in MARL, the novelty of the Switcher-Adversary formulation, and the strength of our empirical results and theoretical guarantees.

**A shared misconception** across reviews is that MARTA resembles prior adversarial or diagnostic frameworks ([A], [B], [1], [2]).
In fact, these works evaluate or attack policies, or model strategic adversaries.

MARTA instead produces robust cooperative policies under true actuator malfunctions—a fundamentally different robustness axis.

Several reviewers overlooked that MARTA is plug-and-play: it robustifies QMIX, VDN, MAPPO and MADDPG without architectural modification. This property is not shared by any cited baselines.

Across reviews, there was underappreciation of MARTA’s theoretical contributions:
(i) a new fault-switching Markov game with budgeted failure dynamics,

(ii) a contraction-based proof of existence and uniqueness of the minimax value,

(iii) policy convergence guarantees even with function approximation.

None of the referenced works provide these results for fault-inducing controllers.

Reviewers requested clarity on modelling choices (single-fault regime, fixed Adversary); these will be addressed with added discussion and extended experiments (including MARTA-B).

Overall, MARTA provides the community with the first unified, theoretically grounded and practical mechanism for training multi-agent policies robust to hardware-like malfunctions, addressing challenges that prior adversarial MARL methods do not cover.

[A] [2] Li, Simin, et al. “Byzantine Robust Cooperative Multi-Agent Reinforcement Learning as a Bayesian Game.”
ICLR 2024.

[B] Zhou, Ziyuan, and Guanjun Liu. “Robustness Testing for Multi-Agent Reinforcement Learning: State Perturbations on Critical Agents.”
arXiv:2306.06136 (2023).

[1] Phan, Tom, et al. “Learning and Testing Resilience in Cooperative Multi-Agent Systems.” AAMAS 2020.

---

### Author Response · Authors · 2025-11-30
**Updates in response to Reviewers' Comments and Requests**

We again thank all participants in the review process. We have addressed the reviewers' comments in our updated manuscript which now includes new experiments, ablation studies and detailed explanations that we believe address all the comments from the reviewers.

Many concerns from the reviewers relate to our empirical evaluation and some missed points about our theoretical contributions in the paper. We have now **highlighted the novel theoretical contribution to MARL robustness** that our paper including elaborated explanations of our proofs and vast differences to existing works. We have **added a comprehensive suite of experiments comparing against existing fault tolerant and robust MARL methods**.

**All updates to our paper (uploaded) are highlighted in red.** We detail the specific updates in response to the comments from each reviewer below:



**Clearer theoretical guarantees and convergence justification** [**Reviewer 1**]

Specifically, we have:

•	Added explicit contraction-based argument for the switching-augmented Bellman operator, including monotonicity and strict contraction lemmas and fixed-point reasoning via Banach theorem.

•	Formal clarification that equilibrium policies correspond to Markov Perfect Equilibria under malfunction dynamics.

•	See updated version: Section 3, proof sketch around Theorem 1 and Proposition 1, pp. 5–6, lines 261–264.

**Clarified fault model and adversary role** [**Reviewer 1**]

Specifically, we have:

•	Expanded explanation of how the Adversary defines the fault policy distribution, distinguishing random-fault and worst-case settings.

•	More precise definition of the Switcher’s control policy and its interaction with Adversary policies.

•	See updated version: lines 154-161 and Section 2.1 “Details on Architecture”, pp. 5–6, lines 223–228.

  **Added more experimental diversity and robustness evaluation** [**Reviewer 2**]

Specifically, we expanded ablation studies on Switching Controls vs random triggering and added more explicit commentary on performance degradation without learned switching. Specifically, we have:

•	Added an additional ablation study varying the malfunction probability $p$, demonstrating MARTA’s robustness under varying malfunction probability $p$. See Figure 7, Section A in the Appendix.

•	Added an additional study describing the relationship trade-off between return and malfunction robustness (w.r.t the collision rate).

•	Introduced clearer experimental axes: who malfunctions, fault policy, trigger rate p, switch cost c, and alignment.

•	See updated version: Section 5, pp. 7–8, lines 378–431.

•	Additional discussion of switch cost c, malfunction probability p, and agent count N.

•	See updated version: Ablation Studies, pp. 10–11.

  **Added stronger empirical evidence against robust baselines** [**Reviewer 2**]

Specifically, we have:

•	New comparison against M3DDPG and introduction of MARTA-MADDPG to show compatibility with adversarial MARL backbones.

•	New comparison against EIR tested in two cases: 1. the EIR training setting, namely a single agent Byzantine malfunctions 2. any agent can fail. The results show in their setting (1) the MARTA enhanced baseline has comparable performance and in the more general setting (2) the MARTA enhanced baseline strongly outperforms EIR.

•	Demonstrated that MARTA improves performance even over an already robust baseline.

•	See updated version: Section 5, Figure 5, pp. 9–10, lines 468–485.



 **Added detailed discussion on practical overhead and deployment feasibility** [**Reviewer 3**]

•	Added explicit computational overhead analysis, clarifying that MARTA introduces only modest training-time cost and negligible execution overhead.

•	See updated version: Computational Overhead subsection, p. 10.

**Broader positioning in related work** [**Reviewer 3**]

Specifically, we have:

•	Expanded Related Work section to better differentiate MARTA from shielding, constrained MARL, and adversarial regularisation approaches.

•	Added a discussion of scalability benefits and plug-and-play design as a robustness layer.

•	See updated version: Section 6, pp. 10–11.

---

### Meta-Review · Area_Chair_4p3L · 2026-01-06

**Summary:**

The paper proposes Multi-Agent Robust Training Algorithm (MARTA), a plug-and-play framework designed to improve the fault tolerance of Multi-Agent Reinforcement Learning (MARL). It introduces a Switcher to decide when and where to trigger malfunctions and an Adversary to control faulty agents in a budgeted adversarial game.

While reviewers recognized the importance of the problem and the novelty of the Switcher-Adversary formulation, the reviewers raised significant concerns regarding the paper’s technical novelty, the simplicity of the evaluation environments, and a potential ethical issue.

While the authors provided rebuttals and promised additional experiments (MARTA-B, M3DDPG comparisons), the core concerns (i.e., technical novelty and empirical depth) remain significant. A rejection is recommended at this stage.

**Reviewer Concerns:**

Concerns still outstanding:

1 Reviewer YpZz flagged that Appendix E appeared nearly identical to several previous publications.
Besides, the Reviewer PSdH noted that several works addressing similar attack vectors and faulty agent scenarios were uncited or not compared against empirically.

2 Concerns regarding the readability of proofs, the justification for "switching controls" vs. "impulse control", and how the minimax equality was proven

3 The tested domains (Traffic Junction, LBF, MPE) were too simple compared to standard benchmarks like SMAC or MAMuJoCo used in cited works. The claim that MARTA scales better than shielding methods was viewed as unvalidated, given experiments were limited to 6 agents while baselines cited handled over 1,000.

**Reviewer Scores:**

1 Reviewer YpZz remained skeptical after rebuttal regarding the lack of comparison to robust baselines and the proof of value existence.

2 Although most concerns are addressed, Reviewer ry1d still feel the "why" behind modeling choices and the title/abstract misalignment remained unresolved.

3 Reviewer PSdH maintained that the theoretical contribution was incremental and that the setting was a straightforward instantiation of known Markov games.

---

### Decision · Program_Chairs · 2026-01-26

Reject